



**Previous integrated or organic farming affects productivity**
**and ecosystem N balance rather than fertilizer $^{15}$N allocation to**
**plants and soil, leaching, or gaseous emissions (NH$_3$, N$_2$O, and**
**N$_2$)**
Fawad Khan [1], Samuel Franco Luesma [2], Frederik Hartmann [3], Michael Dannenmann [1], Rainer
Gasche [1], Clemens Scheer [1], Andreas Gattinger [3], Wiebke Niether [3], Elizabeth Gachibu Wangari
[1], Ricky Mwangada Mwanake [1], Ralf Kiese [1], Benjamin Wolf [1],
[1] Institute of Meteorology and Climate Research, Atmospheric Environmental Research (IMKIFU), Karlsruhe
Institute of Technology (KIT), Garmisch-Partenkirchen, 82467, Germany.
[2] Department for Environment, Agricultural and Forest Systems, Agri-Food Research and Technology Centre of
Aragon (CITA), 50059 Zaragoza, Spain.
[3] Chair of Organic Farming with focus on Sustainable Soil Use, Justus Liebig University, Karl-Glöckner Str. 21C,
35392 Giessen, Germany.
*Corresponding to*: Dr. Benjamin Wolf (benjamin.wolf@kit.edu)





**Abstract**
Legumes in crop rotations are considered an ecological intensification management practice to reduce nitrogen
(N) losses to the environment. However, studies on N allocation and loss on adjacent sites with the same
pedoclimatic conditions but different management histories, i.e. organic farming (OF) with frequent legume
cultivation and occasional organic fertilizer input, compared to integrated farming (IF) with synthetic and organic
fertilizers, have remained scarce. Here, we quantified field N losses (ammonia, nitrous oxide, dinitrogen, and
nitrate leaching), total N balances, and $^{15}$N labelled cattle slurry allocation to soil and plants of two adjacent sites
over a two-years cropping sequence. While IF had resulted in significantly higher pH and soil organic carbon and
N content, the emissions of ammonia, nitrous oxide and dinitrogen after cattle slurry application as well as nitrate
leaching were not significantly different across the two farming techniques. Ammonia losses were low for all
cultivation periods, indicating that drag hose application and manure incorporation successfully mitigates
ammonia emissions. High $^{15}$N fertilizer recovery in plants and soil, along with a low share of unrecovered $^{15}$N
agreed well with the low directly measured N losses. On average, $^{15}$N recovery was lower for OF (85% versus
93% in IF), likely due to unaccounted $N_2$ emissions which could only be measured within two weeks after fertilizer
application, but the high spatial variability of $^{15}$N recovery may have turned this difference insignificant.
Significantly higher harvest biomass N for IF demonstrated that management history affected productivity through
increased soil organic matter mineralization. Due to the higher productivity, the cumulative N balance across all
cultivation period was neutral within the limits of the measurement uncertainty for IF (-8 ± 15), indicating an
optimized N management. For OF, the N balance across single cultivation period ranged from -19 to 41 kg N ha$^{-1}$,
thus, the observations of a single cultivation period were inconclusive. The cumulative positive N balance (48 ±
14) across all cultivation periods for OF suggests that more frequent organic fertilizer additions could increase soil
N (and carbon) stocks, and finally improve yield. However, the positive N balance, coupled with lower $^{15}$N
recovery for OF, also points to a higher likelihood of unaccounted N losses, which would, in turn, slow down the
accumulation of soil N and C over time.
**Keywords:** $^{15}$N tracing, $^{15}$N gas flux, ecosystem nitrogen losses, nitrogen balance, farming systems
comparison, green rye**,** silage maize, perennial ryegrass
**1. Introduction**

42   The excessive and inadequate use of nitrogen (N) fertilizers in agriculture has led to N pollution

worldwide (Good and Beatty, 2011; Cárceles et al., 2022; Pomoni et al., 2023) since gaseous compounds such as
ammonia ($NH_3$), nitrous oxide ($N_2O$) and nitric oxide (NO) are released from agricultural fields into the
atmosphere and dissolved nitrate ($NO_3^-$) enters water bodies through leaching and surface runoff. Adverse effects
of N loss to the environment include eutrophication, biodiversity loss, global warming and air pollution (Liu et al.,
2022; Abdo et al., 2022).

48   To reduce N losses, the 4R nutrient stewardship concept is one of the most prominent approaches

providing advice on the right fertilizer source, at the right rate, at the right time, and at the right place (Bryla, 2011;
Fixen, 2020; De et al., 2024; Nigon, 2024). The adoption of this concept has led to improved nutrient use
efficiency, increased crop yields, and reduced N loss (Surekha et al., 2016; Snyder, 2017; Costa et al., 2020). As
a complementary strategy, legume cultivation is considered an "agro-ecological" intensification practice because



it reduces reliance on external inputs, such as synthetic N fertilizers and associated fossil fuel consumption, due to
their nitrogen-fixing ability. Simultaneously, it enhances biodiversity, improves soil fertility, and their provision
of nutrient-rich food (Reza and Sabau, 2022). Similarly, residue return and the combined use of organic and
mineral fertilizers further enhance soil organic matter, improve soil structure, retain N in the soil, and increase N
availability to plants (Gardner and Drinkwater, 2009; Iheshiulo et al., 2024; Khan et al., 2024). While these
strategies offer clear benefits, studies on arable land that has been managed long-term according to these practices
and which assess balances, allocation, and losses of fertilizer N have remained scarce.

The allocation of N with in the soil-plant system can be determined using the $^{15}$N tracing method which

utilizes fertilizers containing the rare isotope nitrogen-15 (Kramer et al., 2002; Heng et al., 2014; Chalk, 2015).
This method has been successfully applied in arable and grassland systems (Kramer et al., 2002; Zhang et al.,
2012; Quan et al., 2020; Pearsons et al., 2023; Dannenmann et al., 2024). A comprehensive meta-analysis by
Gardner and Drinkwater (2009), which reviewed 217 field-scale $^{15}$N tracing studies, along with a recent global
meta-analysis by Xu et al. (2024) of 79 studies on arable land, showed that the $^{15}$N tracing approach gives useful
information on the ability of management practices to improve N retention and N use efficiency. However, this
method exclusively addresses the allocation of N in soil and plants, while neglecting to account for N losses in the
form of ammonia ($NH_3$), nitrate ($NO_3^-$), nitrous oxide ($N_2O$), and dinitrogen ($N_2$). Nevertheless, such information
is beneficial for fostering effective N management and providing an accurate assessment of N inputs and their
environmental impacts (Zhou et al., 2016). Among the different N loss pathways, the emission of $N_2$ is the least
known since the huge background of atmospheric $N_2$ complicates the determination of the – in comparison – small
soil $N_2$ emissions. The only method for in-situ measurement of $N_2$ is the $^{15}$N gas flux method ($^{15}$NGF) so that
consideration of this loss pathway together with the more easily accessible pathways enhances our understanding
of N allocation in agroecosystems (Kulkarni et al., 2017; Friedl et al., 2020; Dannenmann et al., 2024).

Previous $^{15}$N tracing studies have focused on mineral fertilizers, so there are only a few studies using $^{15}$N-

labelled cattle slurry on arable land (Paul et al., 1995; Buchen-Tschiskale et al., 2023). In addition, there is a lack
of multi-year $^{15}$N tracing studies in current literature directly comparing adjacent farming sites with similar pedo-
climatic conditions but different long-term management histories. Specifically, this includes comparisons between
organic farming (OF), which incorporates frequent legume cultivation and occasional organic fertilizer inputs in a
crop rotation, and integrated farming (IF), which received synthetic fertilizers in combination with organic
fertilizers in a crop rotation.

In this context, the aims of this study were (i) to quantify in-situ N losses ($NH_3$, $N_2O$, $N_2$, and $NO_3^-$

leaching) (ii) to determine $^{15}$N fertilizer recovery in soil and plant; (iii) to calculate field N balances; (iv) compare
the results of (i) to (iii) obtained from two adjacent sites (OF, IF) and (v) relate the differences to the effect of
management history on soil properties.
**2.  Materials and Methods**
**2.1 Study site and historical management and initial soil sampling**

The study was conducted in 65618 Selters, Germany (50°21'28.8"N, 8°15'47.4"E, elevation 310 m a.s.l.),

where the average annual temperature and precipitation amount is 9.3 °C and 655 mm, respectively. There are two
adjacent sites, each approximately 1.0 ha in size, which were selected due to their differing long-term management
histories, identical pedo-climatic conditions, and access to mains power. The organic farming (OF) site was





managed organically for the past decade, with an emphasis on reducing external N inputs so that the majority of

the N input is generated via legumes belowground biomass through biological $N_2$ fixation in the crop rotation.

Periodically, N supply in OF was supplemented with organic cattle slurry (Khan et al., 2024, Table A1). The

integrated farming (IF) site was managed under integrated farming practices, aiming at increasing soil organic

carbon (SOC) through the combination of synthetic and organic fertilizers (Table A1). For the period 2012-2020,

on average, IF received 410 kg C ha$^{-1}$ y$^{-1}$ and 184 kg N ha$^{-1}$ y$^{-1}$, while OF received 200 kg C ha$^{-1}$ y$^{-1}$ and 23 kg N

ha$^{-1}$ y$^{-1}$ (Table A1). The basic soil preparation with regular plowing (20 cm plow depth) and seedbed preparation

was the same at both sites.

To assess the differences in soil physicochemical properties resulting from historical management

practices, a soil sampling campaign was carried out prior to the field trial at four randomly selected locations at

OF and IF sites. At each location, a soil profile was uncovered and sampled at three depths (0–10 cm, 10–30 cm,

and 30–60 cm). The physiochemical analyses included pH, SOC, total nitrogen (TN), soil organic carbon to total

nitrogen ratio (C:N), bulk density (BD), and texture analysis. Soil pH was determined using a pH metre (Metrohm,

Inolab 7310, wtw, Germany) after the soil samples were diluted in distilled water (1:5 soil to water, Dannenmann

et al., 2006). To determine the SOC and TN content, soil samples were ground in a mixer mill (Retsch, MM301,

Haan, Germany), sealed in a tin capsule, and subjected to isotope ratio mass spectrometry analysis (IRMS: Delta

Plus XP; Thermo, Bremen, Germany) according to the protocols of Dannenmann et al. (2016) and Couto-Vázquez

et al. (2020). For BD, additional soil samples were collected using a soil core cutter with a volume of 100 cm³ and

oven-dried at 105 °C for 24 hours (Khan et al., 2020). For texture analysis, soil samples were randomly collected

from ten different locations in each site, pooled, sieved, air-dried, and analysed by a commercial laboratory

(AGROLAB, Agrar GmbH, Sarstedt, Germany).

Due to different management histories, pH, SOC and TN were significantly lower on OF while bulk

density was significantly higher for OF in 0 to 10 cm depth (Table 1). The soil of both sites has a silty loam texture

with almost identical percentages of silt, sand, and clay.

**Table 1:** Soil properties of initial sampling from the organic farming (OF) and integrated farming (IF) sites. All values are given as mean ± standard error (SE, n=4), except for C:N and texture analysis values (only the mean value). In the case of texture analysis, NA represents the samples that were not measured at the corresponding depth.

| Site | Depth [cm] | pH | SOC [%] | TN [%] | C:N [-] | BD [g cm$^{-3}$] | Texture [%] | | |
|------|------------|----|---------|--------|---------|------------------|------|------|------|
| | | | | | | | sand | silt | clay |
| OF | 0-10 | 5.43 ± 0.05 a | 1.04 ± 0.12 a | 0.13 ± 0.01 a | 8.00 a | 1.45 ± 0.01 a | 10.20 | 68.30 | 21.50 |
| IF | 0-10 | 6.84 ± 0.07 b | 1.58 ± 0.07 b | 0.18 ± 0.01 b | 8.77 b | 1.36 ± 0.02 b | 8.20 | 67.10 | 24.70 |
| OF | 10-30 | 5.33 ± 0.06 a | 0.84 ± 0.19 | 0.11 ± 0.02 | 7.63 | 1.43 ± 0.03 a | 11.60 | 65.10 | 23.30 |
| IF | 10-30 | 6.34 ± 0.25 b | 1.08 ± 0.08 | 0.14 ± 0.01 | 7.71 | 1.32 ± 0.04 b | 9.20 | 68.30 | 22.50 |
| OF | 30-60 | 5.90 ± 0.08 a | 0.37 ± 0.03 | 0.06 ± 0.01 | 6.19 | 1.47 ± 0.05 | NA | NA | NA |
| IF | 30-60 | 6.70 ± 0.09 b | 0.62 ± 0.28 | 0.05 ± 0.01 | 11.24 | 1.41 ± 0.05 | NA | NA | NA |

The letters (a, b) specify the significant difference (p < 0.05) between OF and IF sites

## 2.2 Agricultural management during field experiment

The trial period spanned from October 2020 to the end of September 2022, during which fertilizer amount,

type, and crop cultivation were identical on both sites to avoid bias in N loss and allocation due to fertilizer amount.

Using a seed drilling device, 300 kernels per square metre of green rye were sown at a depth of 3 cm with a row



distance of 15 cm in October 2020. On April 8th, 25 m³ ha⁻¹ (88 kg N ha⁻¹, 750 kg C ha⁻¹) of cattle slurry was
applied as a top-dressing application for green rye using a drag hose. Green rye was harvested on May 10th, 2021.
On May 31st, 2021, 20 m³ of cattle slurry (74.8 kg N ha⁻¹, 600 kg C ha⁻¹) was applied as a pre-sowing fertilization
for silage maize. It was incorporated with one pass of a disc harrow followed by one pass of a rotary tiller.
Subsequently, silage maize was sown at a plant density of 90,000 plants ha⁻¹ with a row spacing of 75 cm. Silage
maize was harvested on September 30th, 2021. In October 2021, perennial ryegrass was seeded on both sites at a
density of 45 kg ha⁻¹ with 12.5 cm row spacing. On March 29th and May 30th, 2022 cattle slurry was applied using
a drag hose at a rate of 25 m³ (86 kg N ha⁻¹) and 20 m³ (65 kg N ha⁻¹), respectively. Perennial ryegrass was
harvested on May 28th, July 5th, and September 22nd, 2022.
**2.3 Experimental units and associated measurements**
Since several parameters were quantified, an overview of the associated experimental units is given here
while the detailed method is presented in the dedicated sections. Crop yield was determined on macroplots (4 m
by 4 m), 4 replicates at each site. Dinitrogen and $N_2O$ emissions were measured on mesocosms (0.16 m diameter)
with 6 mesocosms at each site. Recovery of $^{15}N$ in soil and plants was measured on microplots (0.30 m by 0.75 m)
with 4 replicates at each site. Ammonia emissions were determined using semi-open chambers (0.16 m diameter)
with 4 replicates at each site and $NO_3^-$ leaching was determined using self-integrated accumulative (SIA) collectors
which were buried on both sites at four locations in 1 m depth, with three replicates per location for each cultivation
period.
**2.4 Crop biomass yield**
At each harvest, aboveground biomass was harvested in each of the macroplots by harvesting the whole
area using a plot harvester (Haldrup, 1.5 m harvesting width). Biomass dry matter was determined in subsamples
from each macroplot by oven-drying at 60 °C for several days until a stable mass was reached. To determine total
biomass N content, samples were milled and analyses using an elemental analyser (DIN/ISO 13878:1998,
Elementar Analyser system GmbH, Langenselbold, Germany).
**2.5 Determination of dinitrogen emissions from soil-plant mesocosms**
To determine $N_2$ emission, the $^{15}N$ gas flux method ($^{15}NGF$) was applied where $N_2$ emission is calculated
from the isotopic compositions of $N_2$ in atmospheric background and in the headspace of a soil chamber that
previously had received $^{15}N$ labelled fertilizer (Hauck, 1958; Mulvaney et al., 1986; Spott et al., 2006). The
mesocosms were managed in the same manner as the field plots. To this end, six mesocosms from OF and IF were
collected in steel cylinders (0.26 m height and 0.16 m diameter) a month before fertilization and buried in a trench
at the site. For the target application rate of 88 kg N ha⁻¹, cattle slurry was properly mixed with $^{15}N–NH_4^+$ and $^{15}N$-
urea to obtain an enrichment of 85at% (Table A2). The $^{15}N$-labelled cattle slurry was applied to each mesocosm
during green rye cultivation, by simulating a drag hose application. In the silage maize cultivation period, the target
application rate was 74.8 kg N ha⁻¹, and labelled slurry was applied by mimicking drag hose application, and
incorporated into the soil using manual tillage, following the legal regulation in Germany. Subsequently, corn
seeds were planted. In the same way, labelled cattle slurry, with 85at% was prepared for the perennial ryegrass
first (86 kg N ha⁻¹) and second (65 kg N ha⁻¹) application and applied by simulating a drag hose application.



For the $N_2$ gas measurements, 0.16 m high chambers were tightly affixed to the mesocosms using a steel
tension clasp, and gaseous samples were collected using a syringe directly after fixing the chamber and after 2
hours chamber closure time. The syringe was flushed three times with chamber headspace air and 20 ml was
transferred to a pre-evacuated 12 ml exetainer (Model 837W, Labco Limited, United Kingdom), that had been
flushed with helium gas three times. Gas samples were collected daily from day D1 to D7, and additionally on
days D9, D11, D13, D21, and D28, following fertilization.
Isotope ratios of $N_2$ and $N_2O$ in the exetainers were determined using an isotope ratio mass spectrometer
(IRMS: Delta Plus XP; Thermo, Bremen, Germany). These isotope ratios were used to determine the enrichment
of the denitrifying N pool (Arah, 1997; Stevens and Laughlin, 2001b) and, subsequently, the $N_2$ emission according
to Spott et al. (2006). The fraction of fertilizer-derived $N_2$ was determined by calculating the ratio between the $^{15}N$
atom excess percentage of the emitted $N_2$ and the $^{15}N$ atom excess percentage of the applied N fertilizer, following
the procedure given in Yankelzon et al. (2024b).

**2.6 Determination of ammonia losses**

The measurement of $NH_3$ emissions into the atmosphere was conducted using the semi-open chamber
approach, as described by Jantalia et al. (2012), using polyvinyl chloride (PVC) cylinders containing two
polyurethane foams. The foams were pre-soaked in 40-ml 1 M sulfuric acid solution containing 4% (v/v) glycerol
until they absorbed all the solution, and were then placed inside the PVC cylinder on fixtures 5 cm and 20 cm
above the ground. The top foam was used to protect the lower foam disc of $NH_3$ deposition, while the bottom foam
was used to detect $NH_3$ emissions from soil fertilizer N application. Subsequently, the PVC cylinders were placed
onto frames previously installed in the soil of OF and IF. Foam discs were exchanged at daily frequency during
the first week after fertilization and every second day up to a maximum of 14 days of measurement after
fertilization. The $NH_3$ released was captured as $NH_4^+$ on the acidified foams and subsequently extracted using 150-
ml of 2 molar potassium chloride. Extracts of the bottom foam discs were analysed for ammonium ($NH_4^+$) using
an indophenol colorimetry approach based on Kempers and Zweers (1986), and converted to an emission rate in
$NH_4^+$-N by considering the duration of the exposition of the foam disc and the area of the chamber. The quantified
$NH_3$ emissions were considered in the $^{15}N$ balance by calculating the share of volatilized N of the mineral N
amount applied in slurry for each cultivation period.

**2.7 Determination of nitrous oxide fluxes**

Nitrous oxide emissions were determined simultaneously with the $N_2$ emissions by sampling $^{15}N$ enriched
soil plant mesocosms headspace air into 10 ml vials by manual syringe sampling as described above for
dinitrogen fluxes and subsequently determining the mixing ratio using gas chromatographic analysis. The flux
was calculated from the change in $N_2O$ mixing ratio over time during chamber closure using the following
equation (Eq. 1).
$$F = \frac{\Delta C_v \; p \; M \; h}{\Delta t \; A \; R \; T} \tag{1}$$
Where F is $N_2O$ flux ($\mu g \; m^{-2} \; h^{-1}$), p is the atmospheric pressure ($N \; m^{-2}$), M is the molar mass of $N_2O$-N ($\mu g \; mol^{-1}$)
$^{1}$), h is chamber height (m), $\Delta C_v$ is the change in volume mixing ratio (ppm), R is the ideal gas constant ($J \; mol^{-1}$
$K^{-1}$), T is the temperature (K), $\Delta t$ is the duration of the chamber closure (hours), and A ($m^2$) is the surface area of



the mesocosms. The obtained value was used to calculate the product ratio of denitrification by dividing F by the
sum of F and the $N_2$ flux.

**2.8 Determination of nitrate leaching**

Losses in the form of $NO_3^-$ leaching were determined using the SIA method (self-integrating
accumulators, TerrAquat-GmbH; Grahmann et al., 2018). SIAs are absorber materials that were installed at a depth
of 1.0 m in an undisturbed soil section of both sites (OF, IF) and can, thus, collect nitrate during each cultivation
period. In the laboratory, the bound $NO_3^-$ was extracted from the SIAs using a concentrated 2 molar potassium
chloride solution and the nitrate concentration in the extracts was measured using an indophenol colorimetry
approach (Kempers and Zweers, 1986; Grahmann et al., 2018). Nitrate leaching was considered in the $^{15}N$ balance
by calculating the share of leached N of the N amount applied in the labelled slurry for each cultivation period,
implicitly assuming that leaching is due to fertilizer application and occurs within one cultivation period.

**2.9 $^{15}N$ fertilizer tracing on microplots**

**2.9.1 Preparation and application of $^{15}N$-labelled cattle slurry fertilizer**

To determine $^{15}N$ recovery in soil and plants, microplots were established by inserting four steel frames
each (length × width × height: 0.75 m × 0.30 m × 0.15 m; Table A2) into the soil at OF and IF. These microplots
contained two rows of green rye in 2021, following by one row of silage maize in 2021, and two rows of perennial
ryegrass in 2022. For fertilization, cattle slurry was mixed with $^{15}N$-$NH_4^+$ and $^{15}N$-urea to achieve an atomic
enrichment of 10 at%, and the $^{15}N$-labelled cattle slurry was applied manually, mimicking a drag hose fertilizer
application. The amounts of cattle slurry, $^{15}N$-salts and milli-q water for production of spiked slurry were calculated
based on slurry analysis and in a way that the total N content and amount of spiked slurry equalled that of the
original slurry and that the total N amount was 88 kg N ha$^{-1}$ for the green rye cultivation period. Similarly, $^{15}N$–
$NH_4^+$ and $^{15}N$-urea were properly mixed with cattle slurry to achieve a total N amount of 74.8 kg N ha$^{-1}$ for the
silage maize application. Following the legal regulation in Germany, the $^{15}N$-labelled cattle slurry applied before
planting silage maize and incorporated into the soil using a hoe. In a similar manner to green rye and silage maize,
$^{15}N$-labelled cattle slurry was prepared for the perennial ryegrass cultivation period in 2022, yielding 86 kg N ha$^{-1}$
and 65 kg N ha$^{-1}$ for the initial and second application, respectively. The slurry was applied manually, simulating
a drag hose application.

**2.9.2 Calculation of $^{15}N$ recovery in plants and soil**

At the end of each cultivation period, all aboveground biomass was harvested from the microplots, and
belowground biomass was separated from the soil in a vessel. On the date of the first and second perennial ryegrass
harvest in 2022, only aboveground biomass was cut. Plant samples were dried at 60 °C for a week to reach a
constant dry weight, after which they were crushed in a mixer mill (Retsch, MM301, Haan, Germany), and 2 mg
samples were packed in tin capsules for determination of TN, SOC, and the $^{15}N$ atom fraction using an isotopic
ratio mass spectrometer (IRMS: Delta Plus XP; Thermo, Bremen, Germany). The recovery of $^{15}N$ in plants was
calculated from the $^{15}N$ excess compared to natural abundance and the amount of $^{15}N$ applied with the cattle slurry,
with a detailed description being given in Dannenmann et al. (2016).





Similarly, the soil samples from each microplot (0.225 m$^2$) were collected at different depths (0–10 cm,
10–30 cm, 30–60 cm, and 60–90 cm). The soil samples were thoroughly homogenised, sieved (2 mm), and oven-
dried at 60 °C until they reached a constant dry weight. After that, the soil samples were ground in a mixer mill
(Retsch, MM301, Haan, Germany) and packed in tin capsules for measuring SOC, TN, and atom $^{15}$N fraction by
elemental analysis coupled to mass spectrometry (Delta Plus XP; Thermo, Bremen, Germany). In analogy to the
plant recovery, total soil recovery was calculated from the $^{15}$N excess compared to natural abundance and the
amount of $^{15}$N applied with the cattle slurry (Dannenmann et al., 2016).
**2.10  Nitrogen balance**
The N balance was calculated to quantify the difference of N applied to the sites and lost from the sites
(Eq. 2).
$$N\ balance = N_{input} + N_{deposition} - N_{harvest} - N_{loss} \tag{2}$$
Here, $N_{input}$ refers to the N added through fertilizer, $N_{deposition}$ is N from atmospheric deposition (taken from German
Environmental Protection Agency), $N_{harvest}$ accounts for N removed through crop harvest, and $N_{loss}$ includes N lost
via leaching as well gaseous losses in form of $NH_3$, $N_2O$ and $N_2$. To calculate N deposition for specific crop
periods, the annual deposition rate was divided by 365 days and scaled according to the duration (in days) of each
crop cultivation period.
**2.11  Data processing and statistical analysis**
The total $^{15}$N recovery, losses, and N balance were calculated in the Microsoft Excel software program
(Microsoft Office 2019, Microsoft, Seattle, WA, USA). The statistical package of the Social Sciences (SPSS
version 27.0, IBM Crop., Armonk, NY, USA) was used for statistical analysis. Normality was tested using the
Shapiro-Wilk test and based on the result, either the sample t-test or the Wilcoxon test was carried out to test
significant differences at a 95% confidence interval between the OF and IF sites. The soil recovery of $^{15}$N was
calculated as the sum of the recoveries in different soil depths (0–10, 10–30, 30–60, and 60–90 cm). To calculate
cumulative $N_2$ and $N_2O$ emissions, a linear interpolation was made for the days where no measurement was
conducted. Furthermore, OriginPro 2020b (OriginLab Corporation, Northampton, Massachusetts) was used for
illustrations.
**3.     Results**
**3.1 Crop yield**
In the green rye cultivation period, significantly higher harvested aboveground biomass (AGB) and above
ground biomass N (AGB-N) were recorded for IF (Table 2). In contrast, OF showed significantly higher AGB and
AGB-N in the silage maize cultivation period. During the perennial ryegrass cultivation period, the AGB and
AGB-N were significantly higher for IF, and average IF values across the three crops were significantly higher for
AGB-N (20%) but not significantly higher for AGB (13%, Table. 2).



**Table 2:** Average aboveground biomass (AGB) and aboveground biomass-N (AGB-N) with standard error (SE), expressed as absolute dry matter, were reported for the different cultivation periods in organic farming (OF) and integrated farming (IF).

| Period | Treatment | AGB ± SE [Mg ha$^{-1}$] | AGB–N ± SE [kg ha$^{-1}$] |
|---|---|---|---|
| Green rye 2021 | OF | 3.03 ± 0.17 a | 56.84 ± 1.78 a |
| | IF | 5.66 ± 0.21b | 98.47 ± 5.58 b |
| Silage maize 2021 | OF | 9.57 ± 0.53 a | 94.94 ± 3.96 a |
| | IF | 7.14 ± 0.31 b | 75.24 ± 4.21 b |
| Perennial ryegrass 2022 | OF | 8.50 ± 0.13 a | 98.66 ± 3.01 a |
| | IF | 11.13 ± 0.39 b | 132.72 ± 9.86 b |
| Average (2021-2022) | OF | 7.03 ± 0.28 | 83.48 ± 2.92 a |
| | IF | 7.98 ± 0.30 | 102.14 ± 6.55 b |

Lower-case letters (a, b) depict statistically significant differences ($p < 0.05$)

### 3.2 Dinitrogen (N₂) and nitrous oxide (N₂O) emissions

Following $^{15}$N fertilizer applications, increased emissions of total N₂ and fertilizer-derived N₂ were observed within two weeks, with fertilizer-derived emissions accounting for 70%, 63% and 76% for OF and 80%, 55%, 70% for IF of the total emissions during the green rye, silage maize and perennial ryegrass cultivation period, respectively. There was no clear difference in total N₂ flux levels between OF and IF, and flux rates ranged from close to 0 to 1065 µg N m$^{-2}$ h$^{-1}$ (Fig. 1a-q, B1). In contrast, fluxes of total N₂O and fertilizer-derived N₂O for OF were higher after slurry application in the green rye and perennial ryegrass cultivation period, compared to silage maize cultivation period. The fertilizer-derived N₂O emissions accounted for 33, 38, and 33% for OF and 29, 37, and 38% for IF in the green rye, silage maize and perennial ryegrass cultivation period, respectively. However, no clear pattern of elevated total or fertilizer-derived N₂O flux levels was observed for silage maize. Flux rates for total N₂O were low with emissions peaks of approximately 40 µg N m$^{-2}$ h$^{-1}$ in the green rye and silage maize periods and 94 µg N m$^{-2}$ h$^{-1}$ after the first slurry application to perennial ryegrass (Fig. 1c-s), respectively.

The ratio of total N₂O : (N₂+N₂O) showed a similar progression over time for both sites, ranging from 0.01 to 1.0. Since the N₂O and N₂ flux levels were close to the detection limit from day 11 after fertilization, only values before this period were considered in green rye and silage and silage maize cultivation period (Fig. 1e, k). On average, across crops and sites, the N₂O : (N₂+N₂O) was 0.16, and the median amounted to 0.03.

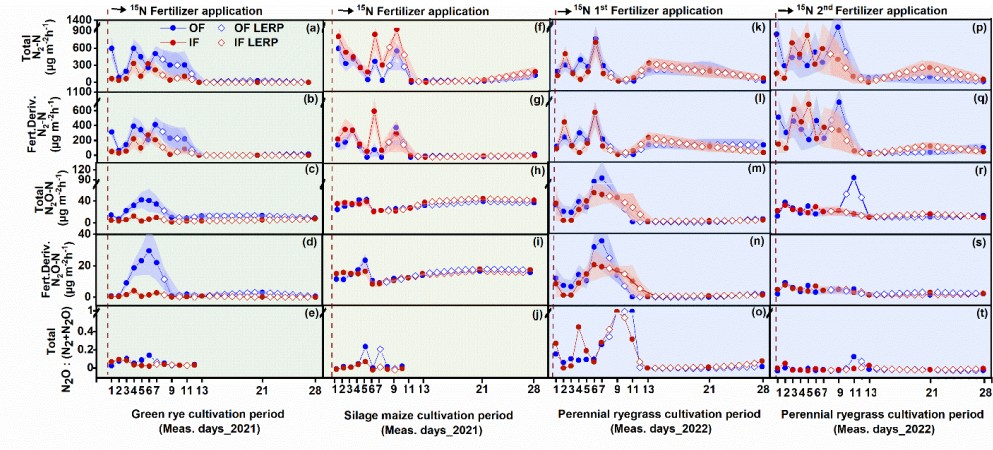



**Figure 1:** Results of the $^{15}N$ gas flux method using mesocosms. From top to bottom, the panels show total $N_2$ flux, fertilizer-derived (fert. deriv.) $N_2$ flux, total $N_2O$ flux, fertilizer-derived (fert. deriv.) $N_2O$ flux, total $N_2O : (N_2+N_2O)$ ratio, following fertilizer application. Measurement periods for fertilization of green rye, silage maize, perennial ryegrass 1, and perennial ryegrass 2 are shown in subplots a–e, f–j, k–o, and p–t, which correspond to green rye, silage maize, the first perennial ryegrass application, and the second perennial ryegrass application, respectively. Filled circles and surrounding bands show mean values and standard errors for OF (blue) and IF (red). Linearly interpolated (LERP) values are marked by white-filled squares.

Cumulative total and fertilizer-derived $N_2$ emissions over the green rye measurement period were significantly higher for OF (1.04 and 0.72 kg N ha$^{-1}$) compared to IF (0.37 and 0.29 kg N ha$^{-1}$, (Fig. 2a, B1). Similarly, cumulative total and fertilizer-derived $N_2O$ emissions were significantly higher for OF (0.10 and 0.04 kg N ha$^{-1}$) than for IF (0.04 and 0.01 kg N ha$^{-1}$, Fig. 2a). During the silage maize cultivation period, cumulative total and fertilizer-derived $N_2$ emissions were with 1.60 and 0.88 kg N ha$^{-1}$ significantly higher for IF than for OF with 0.98 and 0.62 kg N ha$^{-1}$ (Fig. 2b), respectively. For the same period, total $N_2O$ emissions were significantly different between the two sites (OF: 0.26 kg N ha$^{-1}$, IF: 0.29 kg N ha$^{-1}$; Fig. 2b). However, fertilizer-derived $N_2O$ emissions were not significantly different and amounted to 0.10 kg N ha$^{-1}$ (Fig. 2b).

After the first fertilizer application in the perennial ryegrass period, cumulative total and fertilizer-derived $N_2$ emissions were not significantly different, with emissions values of 1.18 and 0.98 kg N ha$^{-1}$ as well as 1.17 and 0.87 kg N ha$^{-1}$ for OF and IF, respectively (Fig. 2c). Similarly, cumulative total (IF: OF: 0.11 kg N ha$^{-1}$, Fig. 2c) and fertilizer derived (IF: OF: 0.04 kg N ha$^{-1}$) $N_2O$ were also not significantly different and comparable for both sites. After the second fertilization of perennial ryegrass, cumulative total (OF: 2.16, IF: 2.20 kg N ha$^{-1}$, Fig. 3d) and fertilizer-derived (OF:1.55, IF: 1.46 kg N ha$^{-1}$) $N_2$ emissions were not significantly different, and $N_2O$ emissions were below 0.05 kg N ha$^{-1}$ for both sites, with no significant differences observed (Fig. 3d). Average emission of total and fertilizer-derived $N_2$ and $N_2O$ for both sites during the green rye, silage maize and perennial ryegrass were comparable and not significantly different (Fig. 3e).

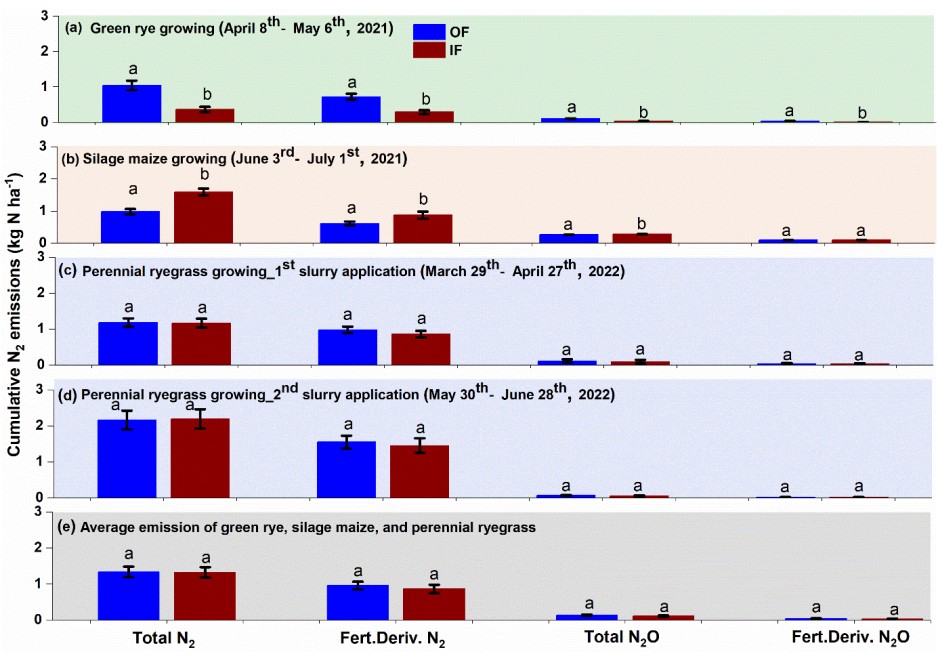



**Figure 2:** Cumulative total and fertilizer-derived $N_2$ and $N_2O$ emission for a measurement period of one month during (a) green rye cultivation period, (b) silage maize cultivation period, (c) perennial ryegrass 1st $^{15}N$-labelled cattle slurry application (d) perennial ryegrass 2nd $^{15}N$-labelled cattle slurry application and (e) average emissions across the crops. Bars with whiskers represent mean values and standard error, respectively.

### 3.3 $^{15}N$ fertilizer recovery

Across the treatments and cultivation periods, total $^{15}N$ fertilizer recovery considering plant, soil and directly measured N loss pathways ranged from 84 to 100% (Fig. 3a-c, Table A3). The highest average recovery for all cultivation periods was typically found in soil (OF: 44%, IF: 46%, Fig. 3d), followed with slightly lower average recovery in plant above and belowground biomass (OF: 33%, IF: 39%, Fig. 3d). Total N losses were dominated by $NH_3$ (0.2 – 8%), followed by nitrate leaching (0 – 5%), and denitrification losses (0.3 – 4%). Average $^{15}N$ allocation in the microplots and to N loss pathways did mostly not greatly differ between IF and OF.

During green rye cultivation, the $^{15}N$ recovery in the soil was 52% for OF and for IF (35%), while during silage maize, it was 59% for IF and 44% for OF (Fig. 3a, b). For perennial ryegrass cultivation the soil $^{15}N$ recovery was slightly higher for IF (44%) than for OF (36% Fig. 3c).

Recovery in the plant followed an opposite trend to soil recovery for green rye and maize, i.e., higher plant recovery was associated with lower soil recovery, except for perennial ryegrass, where both soil and plant recovery were higher for IF (Fig. 3a-c). Recovery from roots was nearly identical for both sites (OF, IF, Fig. 3a-d) across all cultivation period and contributed only marginally to total plant recovery.

Ammonia emission accounted for 3% and 2% of total fertilizer application on IF and OF, respectively during the green rye cultivation period, was less than 1% for silage maize cultivation and amounted to 8% for IF and OF during perennial ryegrass cultivation, with the no significant differences between the both sites (Fig. 3a-c). Similarly, recovery in $N_2O$ was below 0.1% for both OF and IF across all cultivation periods (Fig. 3a-d). Nitrate leaching was in the range of 0 to 5% for the different cultivation periods, but differences between sites were not significant (Fig. 3a-d).

The total $^{15}N$ recovery considering soil, plant, and measured N loss components, was comparable in the green rye (OF: 87%, IF: 84%, Fig. 3a), silage maize (OF: 86%, IF: 95%, Fig. 3b), and perennial ryegrass (OF: 84% and IF: 100%; Fig. 3c) cultivation period. On average $^{15}N$ recovery for all cultivation periods was 85% for OF and 93% for IF (Fig. 3d). The unrecovered portion of $^{15}N$ was 13% and 16% in green rye, 14% and 4% in silage maize, and 17% and 0.1% in perennial ryegrass for OF and IF, respectively, with no significant differences (Table A3). The average unrecovered part of $^{15}N$ for all cultivation periods was 15% for OF and 7% for IF (Table A3), both within the standard error range of 7-16% and 8-11%, respectively, and, thus, within the uncertainty of measurements.



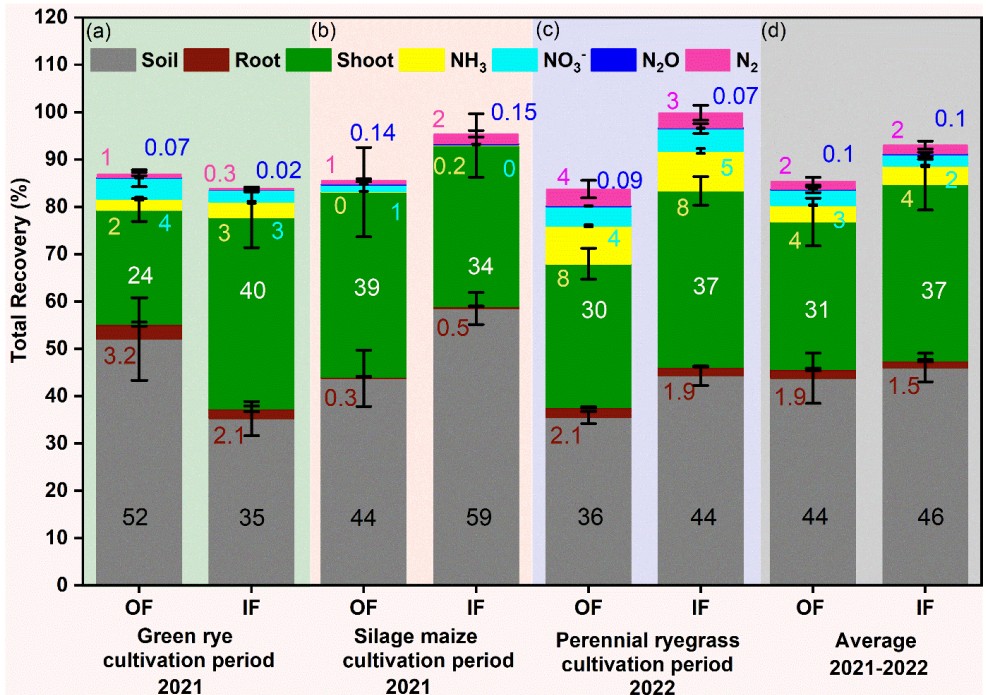

**Figure 3:** Recovery of fertilizer [15]N in soil (0–90 cm depth; grey), aboveground biomass (green), root (vine), ammonia (NH₃, yellow), nitrous oxide (N₂O, blue), dinitrogen (N₂, magenta) and nitrate leaching (NO₃⁻, cyan) for (a) green rye, (b) silage maize, (c) perennial ryegrass cultivation periods and (d) average of all cultivation periods in the organic (OF) and integrated (IF) farming sites. Vertical bars represent mean values, and whiskers represent the standard error (SE) in percentages. All values are presented in rounded decimals to the nearest whole number.

**3.4 Nitrogen balance**

In green rye, silage maize and perennial ryegrass cultivation, 88, 74.8 and 151 kg N ha⁻¹ were applied in the cattle slurry as top dressing in OF and IF (Fig. 4), except for silage maize where it was incorporated. Nitrogen deposition was 1.60 kg N ha⁻¹ for green rye, 2.93 kg N ha⁻¹ for silage maize, and 5.01 kg N ha⁻¹ for perennial ryegrass (Fig. 4a-c), calculated based on data from the German Environmental Protection Agency. In accordance with the [15]N recovery (section 3.3), IF exhibited significantly higher plant N uptake (98 vs. 57 kg N ha⁻¹ in OF, Fig. 4a) for green rye cultivation, while OF showed slightly higher N losses (7 vs. 5 kg N ha⁻¹ in IF) combining NH₃, N₂O, N₂, NO₃⁻ leaching. Consequently, the N balance was positive for OF (26 ± 4 kg N ha⁻¹, Fig. 4a), while IF had a small negative N balance (-14 ± 6 kg N ha⁻¹) with the difference being significant.

In the silage maize cultivation period, OF showed with 95 kg N ha⁻¹ significantly higher N in harvest than the 75 kg N ha⁻¹ of IF (Fig. 4b). Since total N losses were minor compared to yield for both sites (2 kg N ha⁻¹, Fig. 4b), this resulted in a small negative and positive N balance for OF and IF, respectively (-19 ± 4 and 1 ± 5 kg N ha⁻¹), with the difference being significant. Harvest N export during the cultivation of perennial ryegrass was significantly higher for IF than for OF (133 ± 3 and 99 ± 4 kg N ha⁻¹, Fig. 4c), with a comparable N loss both sites.





The N balance was significantly different for both sites, amounting to a surplus of 41 ± 6 for OF and 6 ± 4 kg N

ha⁻¹ for IF (Fig. 4c).

The cumulative harvested N across all cultivation periods of 2021-2022 showed that a significantly higher

amount of N was exported from IF (306 ± 12 kg N ha⁻¹, Fig. 4d) compared to OF (250 ± 10 kg N ha⁻¹). In contrast,

the cumulative N losses were almost similar and not significantly different (OF: 25, IF: 24 kg N ha⁻¹).

Consequently, the N surplus was significantly higher on OF (48 kg N ha⁻¹; Fig. 4d ), compared to IF, which had a

small negative value (-8 kg N ha⁻¹; Fig. 4d).

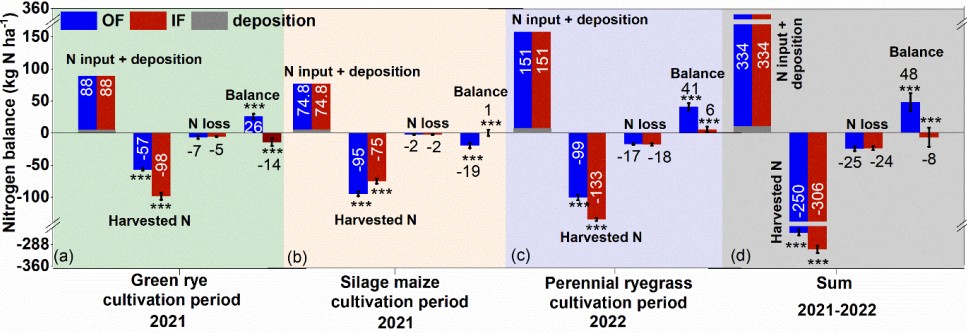

**Figure 4:** Cultivation period N balance for (a) green rye (b) silage maize (c) perennial ryegrass and (d) cumulative N balance for the years 2021 and 2022. N loss combines $NH_3$, $N_2$, $N_2O$ and $NO_3^-$ leaching. The bars with whiskers represent the mean ± standard error values. Organic farming (OF), integrated farming (IF), and soil N deposition are shown in dark blue, red, and grey colour bars, respectively.

## 4. Discussion

### 4.1 Gaseous N loss (ammonia, nitrous oxide, dinitrogen)

A recent summary of experiments on ammonia emissions after slurry application showed that $NH_3$ loss

amounted 10 to 47% of ammoniacal N applied (Häni et al., 2018). The $NH_3$ loss in our study was at the lower end
of the observed range, with 2.4 and 3.2%, 0.2 and 0.2%, and 8.0 and 8.4% (Fig. 3, Table A3) for OF and IF,
respectively, during green rye, silage maize, and perennial ryegrass cultivation periods. While the above range
includes all spreading techniques, in this study manure was applied on every occasion using a technique known to
significantly reduce $NH_3$ emissions. The complete suppression of $NH_3$ emission observed in the silage maize
cultivation period was achieved through the immediate incorporation of the slurry. This agrees with studies
observing reduction of $NH_3$ emission after incorporation and confirms that slurry incorporation is among the most
efficient $NH_3$ mitigation methods if incorporation takes place promptly (Sommer and Hutchings, 2001; Dell et al.,
2011; Sherman et al., 2022). Though manure was applied using the same drag hose applicator during green rye
and perennial ryegrass cultivation, $NH_3$ emissions were variable. Thus, other factors were responsible for the
variability of emissions during green rye and perennial ryegrass cultivation period. Ammonia emission depends
among other factors on ammonium concentration at the manure surface, wind speed, precipitation, solar radiation
and temperature (Sommer and Hutchings, 2001). Consequently, the low temperature and light precipitation on the
date of manure application to green rye reduced $NH_3$ emissions compared to the perennial ryegrass cultivation
period. This agrees with Ni et al. (2015), who reported reduced $NH_3$ emissions following rainfall after fertilization,
and the observation of higher $NH_3$ emissions under warmer and more windy conditions (Sommer et al., 1993; Häni



et al., 2016; Nyameasem et al., 2022; Bleizgys and Naujokienė, 2023). Ammonia emissions of the drag hose
applications (green rye and perennial ryegrass) were much lower than in previous studies using similar trailing
hose application methods, which reported emission rates of 16-45% (Herr et al., 2019; Buchen-Tschiskale et al.,
2023). Consequently, the low levels of $NH_3$ emission in our study were due to efficient $NH_3$ mitigation measures
and the environmental conditions, while the different management histories did not have an effect on the emissions.
In contrast, cumulative $N_2$ emissions were significantly different between OF and IF for the green rye and
silage maize cultivation periods. It is well established that pH has a distinct effect on $N_2O$ and $N_2$ emissions since
at low pH, the synthesis of $N_2O$ reductase, the enzyme catalysing the reduction of $N_2O$ to $N_2$, is inhibited (Russenes
et al., 2016; Zhang et al., 2021). From this perspective, lower $N_2$ emissions would be expected for OF, where the
pH was significantly lower than that of IF (Table 1). However, this was only the case for silage maize cultivation,
suggesting that other factors have contributed to the differences in cumulative $N_2$ emission. For example, lower
plant N uptake on OF compared to IF in the green rye cultivation period may have led to a higher mineral N
availability for microbial processes in the soil. Higher nitrate availability combined with the carbon sources of the
organic fertilizer may have stimulated denitrification, particularly $N_2$ production (Senbayram et al., 2012; Samad
et al., 2016). Thus, the complex interplay of soil N cycle processes like plant N demand, the mineralization-
immobilization cycle controlling N availability, and the effect of environmental conditions on the denitrification
process (Butterbach-Bahl et al., 2002; Chen et al., 2019) led to variable $N_2$ emissions from both sites.
As mentioned above, $N_2O$ and $N_2$ fluxes are closely linked as both are produced and consumed in the
process of denitrification, converting nitrate to nitrite, nitric oxide, nitrous oxide, and finally to $N_2$ (Butterbach-
Bahl and Dannenmann, 2011). For this reason, we expected to observe an increase in both $N_2O$ and $N_2$ emission
after fertilizer application as was reported by Häfner et al. (2021), Herr et al. (2019) and Bizimana et al. (2022). In
this study, we observed a distinct $N_2$ emission following every manure application event, but the $N_2O$ emission
hardly increased for IF during green rye cultivation and for OF and IF during silage maize cultivation (Fig. 2,
Table A3). The other $N_2O$ emission pulses were moderate with emission rates below 140 µg m$^{-2}$ h$^{-1}$ (Fig. 1). The
lack of $N_2O$ peaks after fertilizer application was also observed by Buchen-Tschiskale et al. (2023) and
Dannenmann et al. (2024) and can be explained by conditions conducive to full denitrification, i.e., low oxygen
levels, high soil moisture, good availability of labile carbon sources and high levels of mineral N (Smith and Arah,
1990; Morley et al., 2014; Rohe et al., 2021; Wang et al., 2021). Such conditions can result from the application
of manure, which is a mixture of water, organic substances, and ammonium.
With regard to the magnitude of $N_2$ emissions, few field studies have measured $N_2$ emissions from cattle
slurry application. The level of cumulative $N_2$ emission in this study was between 0.36 and 2.19 kg N ha$^{-1}$ (Fig. 2)
for both sites which is in the same range as the 1.3 to 2.20 kg N ha$^{-1}$ reported by Buchen-Tschiskale et al. (2023)
for an agricultural site receiving 71 kg N ha$^{-1}$ of cattle slurry, and the 0.52 to 0.78 kg N ha$^{-1}$ for a grassland receiving
120 kg N ha$^{-1}$ in form of cattle slurry (Stevens and Laughlin, 2001a). Despite the amount of $N_2$ losses quantified
in this study agreeing with the past studies, the direct field measurement of $N_2$ fluxes in addition to all other
relevant loss pathways did not result in a closed $^{15}$N balance. While some part of the unrecovered $^{15}$N may be due
to an underestimation of $NH_3$ loss, $N_2$ emissions may be underestimated as well (Yankelzon et al., 2024a). One
reason for the underestimation of $^{15}N_2$ is the short coverage of the measurements which is due to the short period
of time during which the isotopic enrichment of the N pool subject to denitrification is sufficiently high so that the
enrichment in the chamber headspace air exceeds the detection limit of the mass spectrometers. Consequently, we
cannot exclude that additional $^{15}$N in $N_2$ is emitted in the months following fertilizer application, particularly during



rewetting events or towards the end of the growing season when plants decrease their water uptake and water
content increases, as observed by Almaraz et al. (2024). Additionally, heterogeneous $^{15}N$ distribution in the
microplot soil resulting from surface or slit application of the slurry and $^{15}N_2$ diffusion and storage in subsoil layers
contributes to the underestimation of $N_2$ flux rates. These complications associated with $^{15}N$ labelling were
discussed in previous studies which indicate that fluxes may be underestimated by up to 30-50% (Vanden Heuvel
et al., 1988; Arah, 1997; Well et al., 2018; Well et al., 2019; Friedl et al., 2020; Micucci et al., 2023; Dannenmann
et al., 2024).

At the same time, however, it must be recognized that the spatial variability of soil properties,

environmental conditions, microbial activity and plant growth leads to a large uncertainty in the determination of
the $^{15}N$ balance and therefore the residual element of the balance may be at least partly due to this uncertainty.
Thus, the apparent mismatch in the N budget may also be due to measurement uncertainties rather than actual N
losses.

The $N_2O : (N_2 + N_2O)$ ratio in this study ranged from 0.01 to 1.00 across both sites using different slurry

application techniques. This aligns with the findings of Fangueiro et al. (2008) who incorporated cattle slurry
fractions (solid and liquid) into the soil, observing that the $N_2O : (N_2 + N_2O)$ ratio varied between 0.32 and 0.73.
Similarly, Dannenmann et al. (2024) observed very low $N_2O$ emissions (0.1 kg $N_2O$-N $ha^{-1}$) from a drag hose
cattle slurry application (97 kg N $ha^{-1}$) under conditions favouring full denitrification, yielding a low $N_2O : (N_2 +$
$N_2O)$ ratio of 0.03.

**4.2 Nitrate leaching**

Nitrate leaching is influenced by several factors, such as nitrate concentration, soil texture, timing of

manure application, meteorological conditions, and cropping system (Van et al., 2006; Maguire et al., 2011;
Wangari et al., 2024). In this study, nitrate leaching rates ranged from 0 to 4 kg N $ha^{-1}$, corresponding to 4 and
3%, 1 and 0%, and 4 and 5% of the applied $NH_4^+$-N (Fig. 3, Table A3) for OF and IF, respectively, during the
green rye, silage maize and perennial ryegrass cultivation period. There was on significant difference between OF
and IF, which indicates that management history had no impact on the $NO_3^-$ leaching. The observed values are at
the lower end of the range of 4-107 kg N $ha^{-1}$ reported for arable cropping systems in a review article (Di and
Cameron, 2002) and the IPCC-based model estimates reported by Eysholdt et al. (2022), averaging 23.1 kg N $ha^{-1}$
for Germany during the reference period of 2014–2016. Among the different crops, nitrate leaching was lowest
for the silage maize cultivation period when manure was incorporated. While preceding studies indicated that
incorporation or injection of slurry increases nitrate concentrations in the soil and increases the risk of N leaching,
differences between application methods vanished at lower slurry manure application rates or if the application
rate was close to N demand (Kramer et al., 2006; Maguire et al., 2011; Dannenmann et al., 2024). The latter was
the case in this study since cumulative aboveground N uptake was close to N application rate (75 to 92%, Figure
4), which may have also caused the low N leaching rates of between 0-1 kg N $ha^{-1}$. Low nitrate leaching associated
with cattle slurry application was also observed in a 4 years study on grassland with trailing hose or cattle slurry
injection of 80 kg N $ha^{-1}$ (Kayser et al., 2015) and by Dannenmann et al., (2024) who reported negligible nitrate
leaching (0.2 kg N $ha^{-1}$) from the cattle slurry application of 97 kg N $ha^{-1}$ to carbon-rich pre-alpine grassland soils
where aboveground N uptake exceeded fertilization rate.



But low leaching rates are not limited to grassland since Buchen-Tschiskale et al. (2023) reported average
nitrate leaching of 4% for trailing hose application or slurry injection of cattle slurry at an application rate of 71
kg N ha$^{-1}$ for arable cropping systems on medium-textured soil, which is close to the values observed in this study.
Thus, the medium soil texture of the sites in this study (Table 1) additionally contributed to the low leaching rates.
Overall, it must be noted that the exact travel time of $NO_3^-$ at the site is unknown and could exceed the
duration of a cultivation period. In addition, total nitrate instead of $^{15}N$-$NO_3^-$ was determined in this study.
Consequently, the observed nitrate leaching for the respective cultivation periods may include some contributions
from preceding fertilizer applications, not fully represent the leaching rate due to the most recent fertilizer
application and overestimate the contribution of $NO_3^-$ leaching to the $^{15}N$ balance.

### 4.3 Recovery of $^{15}N$ from soil and plants

In a single cultivation period, the recovery of $^{15}N$ in plant biomass ranged from 24-39% for OF and 34-
40% for IF, with averages of 31% for OF and 37% for IF (Fig. 4a-d). Soil $^{15}N$ recovery over a single cultivation
period showed a range of 36-52% for OF and 35-59% for IF, averaging 44% for OF and 46% for IF across all
cultivation periods (Fig. 4a-d). To our knowledge, there are only a few studies that used $^{15}N$-labeled cattle slurry
in field experiments on arable land, such as Jensen et al. (2000), who recovered 32% of $^{15}N$ in the aboveground
biomass and 45% in the soil for winter wheat. Similarly, Paul et al. (1995) reported an average recovery of 43%
of applied $^{15}N$-labeled cattle slurry in above- and belowground biomass of corn. In recent work, Frick et al. (2023)
applied $^{15}N$-labeled cattle slurry (produced by feeding a heifer $^{15}N$-enriched ryegrass hay) to a grass-clover system
and observed annual recovery rates of 17-22% in plant biomass and 32-52% in soil. Buchen-Tschiskale et al.
(2023) recovered 32-47% in soil, while plant recovery was between 25-33% using different application techniques
(drag hose and slit injection) for $^{15}N$-labeled cattle slurry to winter wheat, indicating that the recoveries of our
study align well with published data.
Total soil and plant recoveries were highest during the silage maize cultivation period, where manure was
incorporated into the soil. In contrast, recoveries of soil and plant were lower during the cultivation periods of
green rye and perennial ryegrass, where manure application was performed using a drag-hose system. Manure
incorporation distributes N in the soil column, reduces surface N concentration and thus reduces $NH_3$ volatilization
and facilitates microbial immobilization (Sørensen and Thomsen, 2005; Lyu et al., 2024), resulting in overall
recoveries close to 100%. While this trend was not significant in this study, Buchen-Tschiskale et al. (2023)
observed significantly lower $NH_3$ emissions and a significantly higher soil plus plant recovery (79%) and overall
recovery of 99% for slit injection compared to trailing hose application (57 and 78%). Simultaneously,
significantly lower $NH_3$ emissions for the slit injection treatment in Buchen-Tschiskale et al. (2023) and the
incorporation in this study must have resulted in higher availability of mineral N for soil microorganisms which
could lead to increased emission of $N_2O$ or $N_2$. However, in both studies, $N_2O$ and $N_2$ emissions were not
significantly higher, demonstrating that reducing $NH_3$ emissions does not necessarily result in pollution swapping
from $NH_3$ to $N_2O$ or $N_2$ when microbes efficiently immobilize N and prevent direct stimulation of nitrification and
denitrification, which produce $N_2O$ and $N_2$. In general, there was a trend towards lower recoveries in soil and plant
biomass for manure application using a drag-hose which means that other $^{15}N$ loss pathways were more important.
It is well established that the measurement of $NH_3$ emission in the field has remained a challenge and is associated
with large uncertainties (Loubet et al., 2018), suggesting that underestimation of $NH_3$ emissions could explain this
trend. However, since an overall recovery of close to 100% was observed for treatments with $NH_3$ emissions of





8%, i.e., the perennial ryegrass cultivation period of this study and the slit injection treatment of Buchen-Tschiskale
et al. (2023), a significant underestimation of $NH_3$ emission seems unlikely since the underestimated $NH_3$ emission
would imply a recovery greater than 100%. For the same reason, i.e., observation of approximately 100% recovery
for certain treatments, a systematic underestimation of leaching losses is also unlikely. The only measurements
that don't cover the whole cultivation period are those of $N_2O$ and $N_2$, suggesting that underestimation of these N
losses due to coverage of measurements of only a fraction of the whole cultivation period could explain the
unrecovered N losses. Since $N_2O$ emissions are approximately a factor of 10 lower than $N_2$ emissions (Scheer et
al., 2020), $N_2$ emissions may have contributed the main part to the unrecovered losses. In this study, overall
recovery for OF was close to or lower than that of IF, suggesting that for the OF, additional $N_2$ was emitted during
the cultivation period, which could be due to more frequent denitrification events caused by higher soil bulk density
(Table 1, Luo et al., 2000; Hamonts et al., 2013). Furthermore, large error margins in soil and plant recovery rates,
flux measurements, and spatial variation introduce uncertainty, indicating that some of the apparent losses may
also be due to measurement limitations rather than actual N loss.
However, differences in recovery between OF and IF were not significant for single cultivation periods
and the full study period. The different fertilizer application techniques, as well as the different crop types, affect
the interannual variability of recovery rates, which complicates the detection of significant differences on the
interannual scale. In contrast, this suggests for the different cultivation periods that the management history either
didn't affect the allocation of $^{15}N$ to the different components of the $^{15}N$ balance, or that the differences are too
subtle to be determined compared to the spatial variability of soil properties on this specific site and measurement
uncertainty. The highest absolute uncertainties, which eventually also control the overall uncertainty of the
recovery were observed for $^{15}N$ recovery in soil and plants. Though in this study, the area of the microplots was
0.225 m², corresponding to approximately 30 kg of soil for the 0-10 cm layer, only few milligrams of finely ground
soil are eventually used to determine the $^{15}N$ enrichment. Consequently, the excavated soil and plants were
carefully crushed and homogenized in mixing vessels, but the ball mills usually used to prepare the finely ground
material are limited to 15 to 20 g of homogenised material. Since soil aggregates are not entirely destroyed during
homogenization, samples transferred to the ball mill may still show distinct variability which conceals significant
differences between sites. Obviously, increase of replicate amount or analysis of several subsamples (e.g., of soil
of a given layer) could reduce the uncertainty, but only by increasing the already high workload and costs by
several factors. For this reason, our study shows that research into protocols aiming at reducing the uncertainty
arising from incomplete sample homogenisation to the measurement instrument uncertainty by including for
instance mills with much larger sample capacities is required, especially since - to our knowledge - there is no
publication on different homogenisation and mixing protocols for $^{15}N$ balances. At any rate, reduction of
uncertainty for determination of leaching losses, $NH_3$, $N_2$ and $N_2O$ emission is not in view, so that it appears like
the latter loss pathways were not affected by management history.

**4.4 Nitrogen balance and management history**

Previous studies showed that intensive agricultural systems often experience substantial N losses through
gaseous emissions and $NO_3^-$ leaching, which can match or exceed crop N uptake (Ju et al., 2009; Zhou et al., 2016).
In this study, cumulative N losses were not significantly different between OF and IF and can be classified as low
since they amounted to approximately 8 to 10% of plant N uptake. This was due to a combination of the medium
textured soil attenuating $NO_3^-$ leaching and the low $NH_3$ loss resulting from efficient $NH_3$ mitigation measures



(see section 4.1). In contrast, cumulative green rye and perennial ryegrass N uptake were significantly higher on
IF than on OF. Thus, IF was more productive, particularly given that the lower yield during silage maize cultivation
was likely due to delayed herbicide application in response to weather conditions.

Since crop rotation, fertilizer type and fertilizer amount were the same during the two-years experiment

of this study, and climatic conditions were the same for the two adjacent sites during the historical management
from 2012-2020, differences in soil properties due to management must have affected productivity. The
comparison of initial sampling showed that OF soil was significantly more acidic (Table 1), likely due to repeated
legumes cultivation in the crop rotation as legumes take up more cations than anions (Msimbira and Smith 2020,
Chaoui et al., 2023). Additionally, our results show that preceding management resulted in significantly higher
average C and N content down to 30 cm for IF than for OF. On the one hand, the increased N application on IF
during the preceding management of 2012 to 2020 (Table A1) likely boosted above- and belowground biomass
production compared to OF, which resulted in higher C input through above- and belowground plant residues as
well as root exudates on IF which eventually increased SOC levels, in agreement with other long-term studies (Gai
et al., 2018; Böhm et al., 2020). On the other hand, the significantly higher input of a combination of synthetic and
organic fertilizers and returned crop residues on IF (Table A1) has contributed to SOC build-up, and increased
mineralization and microbial biomass (Ramirez et al., 2012; Dai et al., 2017; Tang et al., 2018; Marliah et al.,
2020; Li et al., 2021; Peng et al., 2023; Khan et al., 2024). Consequently, the significantly higher yield can be
related to improved supply of nutrients to the crops between fertilization events through higher mineralization of
organic matter and probably improved water holding capacity (Dai et al., 2017; Manns and Martin 2018; Khan et
al., 2024). In conjunction with the N input through deposition and fertilizer, which was close to the N application
rates for IF from 2012-2020 (average of 184 kg N ha$^{-1}$, Table A1) and in accordance with the national fertilizer
legislation for use of manure (170 kg N ha$^{-1}$ year$^{-1}$), the cumulative and single cultivation period N balances were
neutral within the limits of the measurement uncertainty for IF (-8 ± 15, Fig. 4d). This indicates effective N
management for IF, as a balanced plant N demand with N supply will prevent the mining of soil N and maintain
soil N and C levels through the coupling of the N and C cycles (Zistl-Schlingmann et al., 2020). However, the N
balances across single cultivation period for OF ranged from -19 to 41 kg N ha$^{-1}$ and the cumulative N balance for
all cultivation period was positive (48 ± 14 kg N ha$^{-1}$, Fig. 4d). In a situation in which the boundary of the N
balance is the agricultural field, and N loss pathways (NO$_3^-$ leaching, NH$_3$, N$_2$O, N$_2$) were explicitly determined,
a positive N balance indicates a surplus of N that remains on the field. In other words, this N is available for soil
organic matter build-up, i.e., an increase in soil organic N stocks. This indicates that a more balanced fertilization
approach compared to the management from 2012 to 2020 has the potential to restore C and N levels, which may
increase the productivity of the site in the long-term. However, since the C or N stocks change cannot be
determined on the time scale of 2 years (Küstermann et al., 2013), and – as discussed above – some N loss pathways
do not cover the whole cultivation period and may, thus, be underestimated, it cannot be excluded that the positive
N balance to some degree points towards unaccounted losses. Since positive N balances were observed for OF,
where the $^{15}$N recovery also showed higher unrecovered shares, we assume that not the full N balance can be
attributed to an increase in soil N stocks. In this context, it is noteworthy that crop N export and balance values
from different cultivation periods align with findings from other studies (Thompson et al., 1987; Dell et al., 2011;
Lin et al., 2016; Duncan et al., 2019), which applied cattle slurry using various techniques, including surface
application and incorporation.





Furthermore, the annual variability in N balances of OF was also observed by other studies due to
microbial N immobilization, soil environmental conditions, slurry application technique, crop N export and the
impact of historical management practices (De Jager et al., 2001; Dell et al., 2011; He et al., 2018; Chmelíková et
al., 2021; Winkhart et al., 2022). This underscores the need for repeated N balance measurements and cumulative
N balances to fully capture the effects of different N management practices on N dynamics, ultimately supporting
the development of more sustainable agricultural practices.

**5. Conclusion**

This study is the first to quantify fertilizer N balances, including directly measured emissions of $N_2$, $N_2O$,
$NH_3$, and $NO_3^-$ leaching losses, as well as $^{15}N$ balances and total N balances, over three consecutive cultivation
periods (green rye, silage maize and perennial ryegrass) as affected by either integrated or organic farming history.
Ammonia losses were low due to efficient $NH_3$ mitigation measures, soil texture, and fertilization close
to plant N demand, resulting in low nitrate leaching, irrespective of the management history. Emissions of $N_2O$
were negligible compared to the N balance, and average $N_2$ emissions for all cultivation periods were not
significantly different, but could only be determined for two weeks after fertilizer application. Both OF and IF
practices demonstrated minimal N losses, indicating that both approaches effectively mitigate N leaching and
emissions.
Integrated farming increased productivity by improving plant N supply through higher soil organic matter
and higher rates of mineralization, thereby maintaining a balanced N budget. The positive N balance of organic
farming demonstrates that increased N input may result in soil N and C accumulation, potentially improving
productivity on the long-term. Consequently, a higher level of organic fertilizer additions or a higher share of
residue returns for this specific, legume-cantered N management strategy may be beneficial with regard to
productivity and soil fertility. Finally, this study demonstrates that multi-year $^{15}N$ tracing and N balance studies
are powerful tools to quantitatively assessing the environmental and agronomic impacts of different management
strategies.



**Appendix A: Tables**

**Table A1:** Annual C and N input of organic farming (OF) and integrated farming (IF) sites for the period 2012-2020, i.e., prior to the field experiment. All crops were mowed from both sites with same machinery. Crop residues were returned to IF site but not to OF site. Tillage to a depth of 20 cm was uniformly executed at both sites (OF, IF). In case of missing documentation, gaps were filled based on recommended and typical practice for the area and literature values.

| Year | Organic farming (OF) | | | | Integrated farming (IF) | | | |
|---|---|---|---|---|---|---|---|---|
| | Crop | Input type | kg C ha$^{-1}$ y$^{-1}$ | kg N ha$^{-1}$ y$^{-1}$ | Crop | Input type | kg C ha$^{-1}$ y$^{-1}$ | kg N ha$^{-1}$ y$^{-1}$ |
| 2012 | Silage maize | cattle slurry | 900 | 105 | winter wheat | mineral N | | 180 |
| 2013 | winter spelt | | | | winter wheat | mineral N | | 180 |
| 2014 | red clover & grass | | | | winter wheat | mineral N | | 110 |
| | | | | | | sewage sludge | 795 | 123 |
| 2015 | red clover & grass | | | | silage maize | mineral N | | 110 |
| | | | | | | separated slurry | 837 | 57 |
| 2016 | winter wheat | | | | winter wheat | mineral N | | 180 |
| 2017 | silage maize | cattle slurry | 900 | 105 | winter wheat | mineral N | | 110 |
| | | | | | | sewage sludge | 373 | 60 |
| 2018 | winter wheat | | | | winter rye | mineral N | | 100 |
| | | | | | | separated slurry | 837 | 57 |
| 2019 | alfalfa and grass | | | | winter wheat | mineral N | | 180 |
| 2020 | alfalfa and grass | | | | silage maize | mineral N | | 140 |
| | | | | | | sewage sludge | 848 | 71 |
| Mean | | | 200 | 23 | | | 410 | 184 |




**Table A2:** Preparation of the $^{15}$N labelled manures for different cultivation period.

| Parameters | Green rye cultivation period | | Silage maize cultivation period | | Perennial ryegrass cultivation period 1$^{st}$ application | | Perennial ryegrass cultivation period 2$^{nd}$ application | |
|---|---|---|---|---|---|---|---|---|
| | microplot | mesocosms | microplot | mesocosms | microplot | mesocosms | microplot | mesocosms |
| Area (m$^2$) | 0.225 | 0.16 | 0.225 | 0.16 | 0.225 | 0.16 | 0.225 | 0.16 |
| Replicate/field | 4 | 6 | 4 | 6 | 4 | 6 | 4 | 6 |
| Total unlabeled manure (L) | 4.02 | 0.07 | 3.98 | 0.06 | 4.02 | 0.07 | 3.98 | 0.06 |
| ($^{15}$NH$_4$)$_2$SO$_4$ (g) | 5.75 | 5.61 | 5.06 | 4.93 | 5.75 | 5.61 | 5.06 | 4.93 |
| $^{15}$N urea (g) | 1.33 | 1.30 | 1.17 | 1.14 | 1.33 | 1.30 | 1.17 | 1.14 |
| Milli-Q water (ml) | 534 | 720 | 470 | 633 | 534 | 720 | 470 | 633 |
| Manure + $^{15}$N + mill-Q / replicate (ml) | 503 | 50 | 495 | 50 | 503 | 50 | 495 | 50 |
| Target enrichment level (%) | 10 | 85 | 10 | 85 | 10 | 85 | 10 | 85 |




**Table A3:** Fertilizer $^{15}$N input, $^{15}$N recovery, and unrecovered amounts in green rye, silage maize, and perennial ryegrass
cultivation period.

| $^{15}$N input, loss and recovery | Segment | Green rye_2021 (mean value ± SE) | | Silage maize_2021 (mean value ± SE) | | Perennial ryegrass_2022 (mean value ± SE) | | Average (mean value ± SE) | |
|---|---|---|---|---|---|---|---|---|---|
| | | OF (%) | IF (%) | OF (%) | IF (%) | OF (%) | IF (%) | OF (%) | IF (%) |
| $^{15}$N input | Microplots | 100 | 100 | 100 | 100 | 100 | 100 | 100 | 100 |
| Soil depth | 0-90 cm | 52.10 ± 8.73 | 35.27 ± 3.59 | 43.78 ± 5.95 | 58.53 ± 3.40 | 35.54 ± 1.30 | 44.32 ± 2.05 | 43.81 ± 5.33 | 46.04 ± 3.01 |
| Plant | Shoot | 24.06 ± 2.41 | 40.42 ± 6.36 | 39.00 ± 9.46 | 33.96 ± 6.71 | 30.34 ± 3.28 | 37.16 ± 3.01 | 31.13 ± 5.05 | 37.18 ± 5.36 |
| | Root | 3.17 ± 0.42 | 2.08 ± 0.56 | 0.34 ± 0.07 | 0.46 ± 0.04 | 2.09 ± 0.17 | 1.89 ± 0.02 | 1.87 ± 0.22 | 1.48 ± 0.21 |
| | NH$_3$ | 2.39 ± 0.30 | 3.20 ± 0.20 | 0.19 ± 0.01 | 0.20 ± 0.01 | 8.03 ± 0.20 | 8.45 ± 0.51 | 3.54 ± 0.09 | 3.95 ± 0.19 |
| Nitrogen loss through emissions and leaching | N$_2$O | 0.07 ± 0.01 | 0.02 ± 0.00 | 0.14 ± 0.00 | 0.15 ± 0.00 | 0.09 ± 0.01 | 0.07 ± 0.01 | 0.10 ± 0.01 | 0.08 ± 0.00 |
| | N$_2$ | 0.82 ± 0.36 | 0.33 ± 0.18 | 0.82 ± 0.30 | 2.13 ± 0.70 | 3.55 ± 1.85 | 3.26 ± 1.57 | 1.73 ± 0.84 | 1.91 ± 0.82 |
| | NO$_3^-$ | 4.34 ± 1.79 | 2.56 ± 0.35 | 1.33 ± 0.00 | 0.00 ± 0.00 | 4.09 ± 0.00 | 4.74 ± 1.02 | 3.26 ± 0.60 | 2.43 ± 0.46 |
| Recovery | Sum | 86.83 ± 13.74 | 83.88 ± 11.24 | 85.60 ± 15.80 | 95.43 ± 10.87 | 83.73 ± 6.81 | 99.89 ± 8.19 | 85.42 ± 12.13 | 93.07 ± 10.05 |
| Unrecovered | Input-sum | 13.17 | 16.12 | 14.40 | 4.57 | 16.27 | 0.11 | 14.58 | 6.93 |




**Appendix B: Figure**

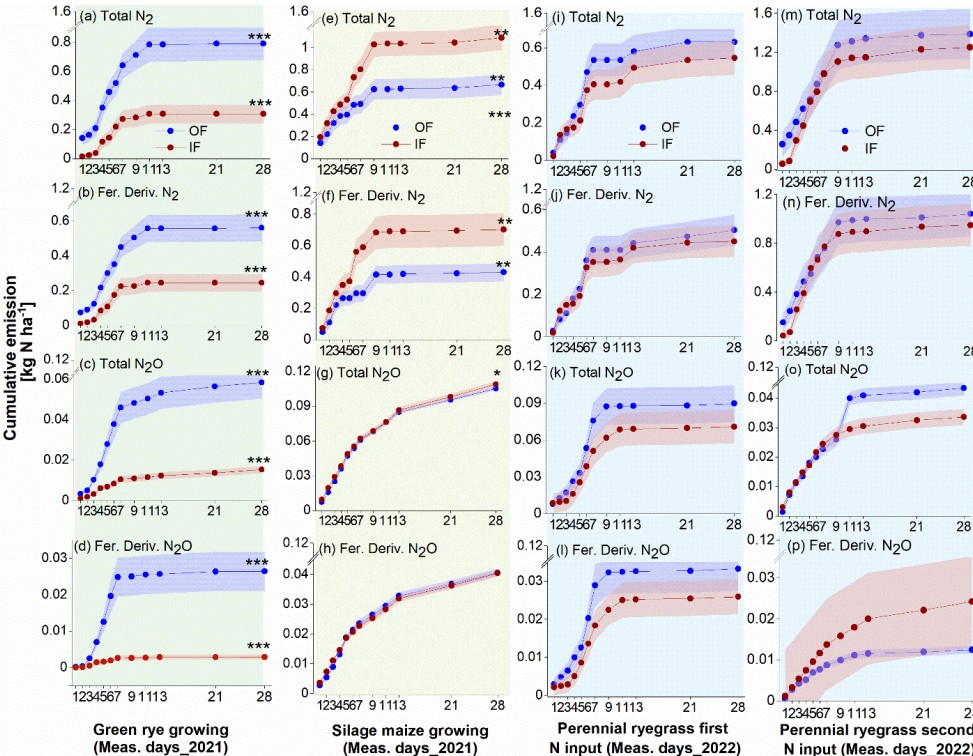

**Figure B1:** The cumulative total and fertilizer-derived emissions of $N_2$ and $N_2O$ across different cultivation period. Subplots (a, b, c, and d) depict the total $N_2$, fertilizer-derived $N_2$, total $N_2O$, and fertilizer-derived $N_2O$ emissions, respectively, during the green rye cultivation period and subplots (e, f, g, and h) show the corresponding emissions for the silage maize cultivation period. For the perennial ryegrass cultivation period, subplots (i, j, k, and l) illustrate emissions of total $N_2$, fertilizer-derived $N_2$, total $N_2O$, and fertilizer-derived $N_2O$ emissions, respectively, following the 1st fertilizer application, while subplots (m, n, o, and p) represent corresponding emissions after the 2nd fertilizer application. Asterisk showed the significant difference (*: $p<0.05$, **: $p<0.01$, ***: $p<0.001$). Symbols and shade represent the mean values with standard error, respectively. The background shading in light green, pale yellow, and light blue represents measurements during the green rye, silage maize, and perennial ryegrass cultivation period, respectively.



**Data availability**

The data used in this manuscript is available from the Karlsruhe Institute of Technology, Garmisch-Partenkirchen, Germany, database. Currently, the dataset can be accessed upon request from the corresponding author and will be made open access upon the final publication of the manuscript.

**Author contributions**

Conceptualization: FK carried out field- and laboratory work, data analysis, and drafted the manuscript. SFL carried out field- and laboratory work. FH carried out field management, data collection and laboratory work. MD, CS and RK supervised, contributed to the study design and revised the manuscript. RG supervised field work. WN carried out the field management and collected data and revised the manuscript. EGW and RMM helped in laboratory work. BW and AG acquired the funding, supervised and revised the manuscript.

**Conflict of interest**

The authors declare no conflict of interest.

**Acknowledgement**

The research was funded by the Federal Ministry of Nutrition and Agriculture (BMEL), Germany, with the grant numbers 2220NR083A, and 2220NR083B.



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
