# Peer review of "Effect of preceding integrated and organic farming on 15N"

_EGUsphere, 2025_

## Author Comment (AC1)

**Author's response to Reviewers' comments #2**

**Manuscript reference number**: MS No.: egusphere-2025-292

**Manuscript title:** Previous integrated or organic farming affects productivity and ecosystem N balance rather than fertilizer $^{15}N$ allocation to plants and soil, leaching, or gaseous emissions ($NH_3$, $N_2O$, and $N_2$)

Dear Editorial Team (egusphere), and Dear Eduardo Vázquez,

We sincerely appreciate your thorough and insightful review of our manuscript. We have carefully addressed your comments and made corresponding revisions to the manuscript. Thank you for giving us the opportunity to improve and resubmit our work. Please find below our detailed responses to each of your points.

| Reviewer #2 |
|---|
| 1) I have reviewed the manuscript entitled "Previous integrated or organic farming affects productivity and ecosystem N balance rather than fertilizer $^{15}N$ allocation to plants and soil, leaching, or gaseous emissions ($NH_3$, $N_2O$, and $N_2$)." The study is highly relevant and has been very well executed using an elegant and robust methodological approach. Furthermore, the manuscript is clearly written, and the results are carefully analyzed and discussed. Therefore, I consider the manuscript worthy of publication, as it contributes significantly to our understanding of how agricultural management practices shape nitrogen cycling.

*__Answer__: We sincerely thank you for taking the time to read and evaluate our manuscript. We are pleased that you consider the manuscript a significant contribution to the understanding of nitrogen cycling in relation to agricultural management practices.* |
| Despite a few minor corrections and suggestions (listed below), my main concern lies in the description and interpretation of the management practices, as well as how the experimental design may limit conclusions about the actual effects of those practices.

*__Answer__: Thank you for your constructive feedback. We have incorporated the minor corrections and suggestions (see below, Minor Comments section) and addressed your primary concern as follows:*
*We revised the Materials and Methods section (Lines 128-129) to better clarify the study's focus on the legacy effects of former management practices. Additionally, we added a new section in the Discussion (Section 4.5 Impact of Experimental Homogenization of management on Legacy Effects) that explicitly addresses how the experimental design limits the interpretation of management effects on nitrogen cycling.*
*We also revised the Conclusion to reflect these limitations and present a more cautious interpretation of the results.* |
| As I understand it, although two contrasting management histories (organic farming, OF, and integrated farming, IF) are compared, both soils were managed identically during the experimental period (2021–2022). Thus, the study does not directly compare OF and IF, but rather the legacy effects of those systems. Although the authors acknowledge this limitation in the Materials and Methods (Lines 121–122) and in other parts of the manuscript (e.g., title, abstract [Lines 17–18], introduction [Lines 78–81], and conclusions [Line 605]), I believe a more explicit discussion is needed on how the homogenization of management during the experiment may have biased or limited the interpretation of OF and IF impacts on N cycling.

*__Answer__: We sincerely appreciate this insightful observation. We agree that our study compares legacy effects of long-term organic (OF) and integrated (IF) management rather than their current practices, as both systems received identical management during the experimental period (2021-2022). While we did acknowledge this limitation in several manuscript sections (as noted), we agree* |

*that a more thorough discussion of its implications was needed. In the revised Discussion we added a section (4.5 Lines 647-661), we now explicitly address which reads as follows:*

**4.5 Impact of experimental homogenization of management on legacy effects**

While the study aimed to investigate the legacy effects of organic farming (OF) and integrated farming (IF) systems on nitrogen cycling, it is important to recognize that both systems were managed identically during the experimental period (2021–2022). This homogenization of management practices, which included uniform crop rotations, tillage methods, and high cattle slurry application, may have limited the ability to draw direct comparisons between OF and IF systems in the experimental phase. As such, the observed differences in N cycling may be more reflective of past management legacies rather than current practices. Additionally, the timing of the crop assessments—green rye immediately after management homogenization and ryegrass two years later—could have further influenced the results, as the green rye was likely more influenced by previous management practices. Furthermore, the high rates of cattle slurry application, typical of IF systems but higher than those generally applied in OF systems, may have masked potential differences between the two systems in terms of nutrient cycling. Finally, the absence of legumes in the experimental crop rotations meant that their role in N cycling could only be inferred as a legacy effect, rather than being directly assessed. These factors should be considered when interpreting the findings, and future studies that include a more direct comparison of management systems with consistent practices and crop rotations, including legumes, would be valuable in further understanding the impacts of OF and IF on N cycling."

For instance, some observed differences across crop periods may result from the time elapsed since management homogenization. Green rye was evaluated immediately after management unification, while ryegrass was assessed two years later. Additionally, during the experimental phase, no inorganic N fertilizer was applied; only cattle slurry was used, and at high rates (136 kg N ha⁻¹ in 2021 and 151 kg N ha⁻¹ in 2022), similar to conventional IF practices, and notably higher than typical rates in organic systems. This might have masked key differences between the management systems. It is also important to note that legumes were not part of the rotation during the experimental phase, meaning their role can only be inferred as a legacy effect, rather than being directly assessed. I strongly recommend including a discussion on these aspects to help readers better understand the scope and limitations of the findings related to management system impacts.

*Answer: Thank you for your comment. We agree with the reviewers and we have addressed this concern in the Discussion (Section 4.5, Lines 647-661), where we added a paragraph discussing the limitations of the experimental design, including the impact of homogenized management practices, the timing of crop evaluations, the high rates of cattle slurry applied, and the absence of legumes in the rotation. This discussion highlights how these factors may have influenced the observed results and how the study primarily reflects legacy effects rather than direct comparisons of organic and integrated farming practices.*

**Minor Comments**

**- L16–17**: The role of legumes in crop rotations is not directly evaluated in this study. I suggest removing or rephrasing this part of the abstract.

*Answer: Thank for the comment. We agree that the role of legumes in crop rotation is not directly evaluated. The sentence is modified and reads as follows (lines 15-16):*

"Ecological intensification strategies in agriculture, including organic fertilization and diversified crop rotations, aim to reduce nitrogen (N) losses to the environment"
* * *
**- L17–20**: The lack of prior studies on this specific comparison does not alone justify the study. Consider strengthening the justification by emphasizing the relevance or potential impact of the findings.

*Answer: Thank you for this valuable suggestion. We agree that the justification for the study should go beyond identifying a knowledge gap. We have added the statement and specifically, we now emphasize that such insights are essential for optimizing nutrient use, reducing environmental impacts, and informing sustainable agricultural practices, which reads as follows (lines 19-21):*

"Understanding how these systems differ in their nitrogen dynamics is essential for improving nutrient management strategies, mitigating environmental impacts, and guiding sustainable agricultural practices."
* * *
**- L54 and L93**: Use consistent terminology, e.g., "symbiotic $N_2$ fixation."

*Answer: Thank you for pointing this out. We have revised the text to use the term "symbiotic $N_2$ fixation" consistently in both Line 54 and Line 93 to maintain clarity and uniformity. The sentences now read as follows (Line 55 and Line 98) .*

"….. associated fossil fuel consumption, due to their symbiotic $N_2$ fixing ability"

"However, nitrogen inputs from symbiotic $N_2$ fixation were…"
* * *
**- L79–81**: Clarify that the study evaluates the legacy effects of previous management practices, not their current application.

*Answer: Thank you for this important suggestion. We have revised the statement to clearly state that the study focuses on the legacy effects of past management practices rather than current management, which reads as follows (Lines 81-85):*

"Specifically, this study assesses how preceding management affects N balance and fertilizer N allocation by comparing sites with different management history. On the organic farming site (OF), legumes had been cultivated frequently, and external N was only occasionally added in the form of cattle slurry, whereas a combination of synthetic and organic fertilizers had been used on the integrated farming site (IF)."
* * *
**- L88**: What does "65618" refer to? Please clarify.

*Answer: Thank you for your comment. This is the postal code, which we have removed as coordinates are given. The sentence now reads (line 92):*

"The study was conducted in Selters, Germany (50°21'28.8"N, 8°15'47.4"E; elevation 310 m a.s.l.), where the average annual temperature and precipitation are 9.3 °C and 655 mm, respectively."

**- L95**: The phrase explaining integrated farming is redundant and unclear, revise for clarity.

*Answer: Thank you for the helpful suggestion. We have revised the sentence to remove redundancy and improve clarity. The updated sentence now reads (Lines 100-101):*

"The integrated farming (IF) site was managed using a combination of synthetic and organic fertilizers, with the aim of enhancing soil organic carbon (SOC) levels (Table A1)."

**- L97–98**: Was there any estimation of nitrogen inputs derived from symbiotic $N_2$ fixation during the previous management phase? Or at least the legume biomass production.

*Answer: Thank you for the comment. Unfortunately, no direct measurements or detailed records of symbiotic $N_2$ fixation or legume biomass production were available for the previous management phase. This limitation is now noted in the revised manuscript in the method section [Lines 98-99].*

"However, nitrogen inputs from symbiotic $N_2$ fixation were not measured prior to ours study. Therefore, these inputs are not included in the reported N input values (Khan et al., 2024, Table A1)."

**-L121–122**: I recommend explicitly noting the potential biases introduced by management homogenization here.

*Answer: Thank you for the insightful comment. We agree that management homogenization, while necessary to reduce confounding effects during the trial period, may introduce potential biases by masking legacy effects of previous land use. To address this, we have added the sentence to explicitly acknowledge this limitation. The sentence reads as follows (Lines 128-129):*

"However, we acknowledge that this management homogenization may mask legacy effects and introduce potential biases in interpreting differences between sites."

**- L133**: Including photographs or diagrams of the experimental setup would improve reader understanding.

*Answer: Thank you very much. We have added details of the experimental design, including all parameters and the number of replicates for both organic and integrated farming.*

[Figure]

Gas sampling

Absorber blocking ambient emissions

Absorber traping manure emissions

$NH_3$ $NH_3$

| Microplots ($^{15}$N recovery) n = 4 per site | Macroplots (crop yield) n = 4 per site | Nitrate leaching (1 m depth) n = 4 per site | Ammonia trap n = 4 per site | Mesocosms ($N_2$ and $N_2O$ emisson) n = 6 per site |

Self-integrated accumulative collecters

**Figure 1:** Schematic representation of the experimental units and associated measurements to assess the N balance and fertilizer N allocation. Microplots were established to assess $^{15}$N recovery from soil and plants, macroplots to measure crop yield, self-integrated accumulative collectors (at 1-meter depth) to monitor nitrate

leaching, semi open chambers with absorbers for ammonia, and mesocosms to quantify gaseous N emissions ($N_2$ and $N_2O$) through gas sampling. The experiment included 4 replicates per site for microplots, macroplots, nitrate leaching, and ammonia traps, as well as 6 replicates per site for mesocosms.

**- L353–354**: "Silage maize" is mentioned twice, please correct.

**Answer:** *Thank you for pointing this out. We have broken the sentence into two parts to improve clarity and avoid redundancy. It now reads (Lines 370-372) :*

"In green rye, silage maize, and perennial ryegrass cultivation, 88, 74.8, and 151 kg N ha$^{-1}$ cattle slurry were applied. For green rye and perennial ryegrass, cattle slurry was applied as top dressing in OF and IF (Fig. 4), while for silage maize, the slurry was incorporated."

**- L402–403**: Clarify "different", in what sense were they different? Which value was higher?

**Answer:** *Thank you for the comment. We clarified this in the following sentence (Lines 420-421):*

"In contrast, cumulative $N_2$ emissions were significantly different between OF and IF for the green rye and silage maize cultivation periods, with higher emissions under OF for green rye and under IF for silage maize."

**- L402–413**: Discuss how differences across managements and crops could be influenced by the time elapsed since management homogenization. Notably, differences were not observed in the last two crops, possibly due to longer homogenization time.

**Answer:** *Thank you for the suggestion. We have added a discussion addressing how differences between managements and crops may be influenced by the time since management homogenization. The discussion reads as follows (Lines 426–432):*

"One such factor could be the time elapsed since management homogenization. The same crop and fertilizer management were applied to both sites starting in October 2020, but legacy effects from the prior organic or integrated farming histories may still have influenced emissions, especially during the early cropping periods like green rye and silage maize. Over time, these legacy effects may have diminished, which could explain the lack of significant differences in $N_2$ emissions observed during the later crop (perennial ryegrass). Similarly, crop-specific factors may have interacted with site history. For example, lower plant N uptake on OF compared to IF in the green rye cultivation period may have led to a higher mineral N availability for microbial processes in the soil. Higher nitrate availability combined with the carbon sources of the organic fertilizer may have stimulated denitrification, particularly $N_2$ production (Senbayram et al., 2012; Samad et al., 2016). In contrast, by the time of the perennial ryegrass phase, soil microbial communities and N cycling dynamics may have adjusted to the unified management, leading to more comparable emissions. Alternatively, lower plant…."

**- L407–411**: The high nitrogen dose applied compared to the previously organic system, could have influenced the observed effects. This should be acknowledged.

**Answer:** *Thank you for this important point. We have acknowledged in the manuscript that the higher nitrogen dose applied to the previously organic system could have influenced the observed $N_2$ emissions and other effects, as follows (Lines 437–442):*

"Additionally, the relatively high N input applied during the experimental period compared to the historically low N input particularly under OF may have intensified the denitrification response, particularly in the early cropping periods. Thus, the complex interplay of historical management, time since management harmonization, crop type, and soil N cycle processes like plant N demand, the mineralization-immobilization cycle controlling N availability, and the effect of environmental conditions on the denitrification process (Butterbach-Bahl et al., 2002; Chen et al., 2019) led to variable $N_2$ emissions from both sites."
* * *
**-L461**: "There was no…"

**Answer:** *Thank you for pointing out. We have corrected the sentence which reads as follows (Line: 491).*

"There was no significant difference….."
* * *
-**L476–479**: This short paragraph should be more clearly connected to the previous one for better flow.

**Answer:** *Thank you for the suggestion. We have slightly revised and integrated the short paragraph to improve clarity and flow, which reads as follows (Lines 495-497):*

"Similarly, comparable $NO_3^-$ leaching values on medium-texture soil were reported by Buchen-Tschiskale et al. (2023), who observed average nitrate leaching of 4% for trailing hose application or slurry injection of cattle slurry at an application rate of 71 kg N ha$^{-1}$ for arable cropping systems."
* * *
- **L479**: As this is a hypothesis, rephrase the sentence to reflect speculative language.

**Answer:** *Thank you for the comment. We have revised the sentence to use more speculative language, now reading (Lines 498-499):*

"Thus, the medium soil texture of the sites in this study (Table 1) may have contributed to the observed low leaching rates."
* * *
- **L599–601**: Consider adding a recommendation for future studies to investigate both legacy and contemporary impacts of agricultural management practices.

**Answer:** *Thank you for the suggestion. We have added the recommendation for future studies which reads as follows (Lines 642-645):*

"To improve our understanding of long-term nutrient cycling, future studies should explicitly address both the legacy effects of past management practices and the direct impacts of current interventions. Such research will be crucial for disentangling time-dependent effects and guiding the design of resilient and sustainable farming systems."

---

## Author Comment (AC2)

**Author's response to Reviewers' comments #1**

**Manuscript reference number**: MS No.: egusphere-2025-292

**Manuscript title:** Previous integrated or organic farming affects productivity and ecosystem N balance rather than fertilizer $^{15}$N allocation to plants and soil, leaching, or gaseous emissions (NH$_3$, N$_2$O, and N$_2$)

Dear Editorial Team (egusphere), and Dear Referee,

We sincerely thank you for taking the time to carefully review our study and for your constructive and valuable feedback. We have revised the manuscript according to the valuable comments and thank you for the opportunity to submit a revised version of our manuscript. Please find below the detailed one-to-one responses to your comments.

| Reviewer #1 |
| --- |
| 1) This is a very important study comparing the N budget and N transformations in organic and integrated farming. 15N tracing method is applied and all the components of N-cycling are analysed , both mineral N and gaseous N forms. This very detailed study allows to almost close the N budget which is a very challenging task and authors manage this exceptionally well.

 *Answer: We thank the Referee for their positive and encouraging feedback. We appreciate the recognition of the comprehensive approach taken in our study, particularly the use of the $^{15}$N tracing method and $^{15}$N gas flux method and the detailed analysis of N cycling components. We are glad that the effort to close the N budget has been acknowledged and valued.* |
| The manuscript is very well prepared, provides the summary of results very clearly, although this is a very complex dataset. I've read this manuscript with pleasure and interest, and I definitely support the publication in Biogeosciences. I only have some minor comments which could strengthen some technical aspects of the manuscript and the data discussion. Especially, to enhance your discussion I suggest to keep it quantitative, since you have quantified all the N fluxes you may quantitatively check all of your discussed assumptions (I give examples in the specific comments). I think it is important since your work is very valuable due to tracing and analysing all of the N budget components and, as far as I know, is probably closest to fully close the N budget.

 *Answer: We sincerely thank the referee for their thoughtful and encouraging evaluation of our manuscript. We are pleased to hear that the complexity and scope of the dataset were clearly conveyed and that the manuscript was found to be both interesting and enjoyable to read. We appreciate the suggestion to further strengthen the technical aspects of the discussion by keeping it more quantitative. In response, we have carefully reviewed the discussion and incorporated additional quantitative comparisons where relevant, particularly where assumptions were made based on the data. We agree that this enhances the clarity and robustness of the conclusions and supports the value of our attempt to comprehensively trace and quantify all components of the nitrogen budget.* |
| **Specific comments** |
| 1. Title – is very complex and difficult to follow, containing a conclusion, which is discussed in the manuscript, but not fully sure,  I would suggest a simplification and more generalisation, like eg. Comparison of N balance in integrated and organic farming: 15N tracing approach with analysis of |

all N-compounds (…) - the title should rather contain a method applied and an aim of the study, not the main conclusion

**Answer***: Thank you for your comment on title. We agree that the title was complex and have revised it to better reflect the aim and methodology of the study, and management history of the study in line with the recommendation. We replaced the previous title with (Lines 1-3):*

"Effect of preceding integrated and organic farming on $^{15}N$ recovery and the N balance, including emissions of $NH_3$, $N_2O$, $N_2$ and leaching of $NO_3^-$"
* * *
2. L32 units are missing for IF (-8 ± 15) – not clear what this value means

**Answer***: Thank you very much for pointing this out. We have added the missing units (kg N ha$^{-1}$) to the value for IF to clarify its meaning. The revised sentence now reads as follows (Lines 32-33):*

"Due to the higher productivity, the cumulative N balance across all cultivation period was neutral within the limits of the measurement uncertainty for IF (-8 ± 15 kg N ha$^{-1}$)…"
* * *
3. L 34 as above N balance (48 ±14).

**Answer:** *We thank the referee for pointing this out. We have added the missing units (kg N ha$^{-1}$) to the value for to clarify its meaning. The revised sentence now reads as follows (Lines 35-36):*

"The cumulative positive N balance (48 ± 14 kg N ha$^{-1}$)…"
* * *
4. L 72-74 "The only method for in-situ measurement of N2 is the 15N gas flux method" – this is not true because also natural abundance isotope analyses of N2O can be used to quantify N2O reduction and hence – calculate the N2 flux (please check: https://bg.copernicus.org/articles/14/711/2017/, https://bg.copernicus.org/articles/17/5513/2020/ ). The method has of course its limitations, but 15NGF also has (e.g. the high detection limit and short time of possible measurements after tracer application).

**Answer:** *Thank you very much for this important clarification. We agree that the $^{15}N$ gas flux method is not the only in-situ approach available for estimating of $N_2$ emissions. As correctly noted, natural abundance isotopologue analyses of $N_2O$ can also be used to infer $N_2$ production via $N_2O$ reduction, as demonstrated by studies such as Lewicka-Szczebak et al. (2017 and 2020).*
*We have revised the statement in the manuscript to reflect this and to avoid overgeneralization. The revised sentence now reads as follows (Lines 73–77):*

"Available methods for in-situ measurement of $N_2$ fluxes, include natural abundance isotope approaches (Lewicka-Szczebak et al. 2017, 2020) and $^{15}N$ labelling (Micucci et al., 2023). The $^{15}N$ gas flux method ($^{15}NGF$) is a well-established approach for quantifying $N_2$ losses and considering this loss pathway together with the more easily accessible pathways enhances our understanding of N allocation in agroecosystems (Kulkarni et al., 2017; Friedl et al., 2020; Dannenmann et al., 2024)"
* * *
6. L 167-168 It would be good to describe the preparation methodology, which peripheral was used, which masses were measured, what was the detection limit and precision of the measurements.

These are very demanding analyses, so these details are necessary. Please add a citation of the preparation method applied.

**Answer:** *Thank you very much for your comment, and we appreciate the suggestion to provide more detailed methodological information. Here we have provided full information (Lines 180-187) :*

"Gas samples were analyzed using an isotope ratio mass spectrometer (Isoprime PrecisION, Elementar UK Ltd., Stockport, UK), coupled to an isoFLOW GHG GasBench (Elementar UK Ltd.). This setup allows for subsampling of gas volumes (30 µL) for $^{15}N\text{-}N_2$ analysis by measuring m/z 28, 29, and 30, and for analysis of $^{15}N\text{-}N_2O$ after cryogenic pre-concentration of $N_2O$ of the remaining vial content by measurement of m/z 44, 45, and 46. Gas handling and preparation followed the protocol described in Arah, (1997); Stevens and Laughlin, (2001b); Spott et al. (2006). In-run uncertainty, i.e., the standard deviation determined from repeated analysis of reference gases for isotope ratios $r^{29}$, $r^{30}$, $r^{45}$ and $r^{46}$ amounted to $1\ 10^{-6}$, $5\ 10^{-7}$, $7\ 10^{-5}$, and $4\ 10^{-5}$, respectively. At the enrichment of the fertilizer added to the mesocosms (85% at), this standard deviation values relate to $87\ \mu g\ N_2\text{-}N\ m^{-2}\ h^{-1}$."
* * *
8. L 325 "52% for OF and for IF (35%)" – the bracket should be removed.

**Answer:** *Thank you very much. We have removed the brackets, and the sentence now reads as follows (Line 342):*

"During green rye cultivation, the $^{15}N$ recovery in the soil was 52% for OF and 35% for IF…"
* * *
9. L 433-443 You discuss the possible N2 flux underestimation as the missing component of your N-balance. It would be interesting to make some estimations with real values to check this theory in practise. Eg. if we assume the 50% of N2 underestimation (as literature data suggest) will this really be sufficient to close the N budget? Just looking at the fluxes, I think it isn't. How large should be the N2 flux really to fill the missing budget? Would this amount be realistic?

**Answer:** *Thank you for this valuable comment. Based on our N balance, the unrecovered N was 15 kg N ha⁻¹ for OF and 7 kg N ha⁻¹ for IF. Our measured N₂ fluxes were approximately 1.0 kg N ha⁻¹ for both treatments. Assuming a 50% underestimation (as suggested in the literature), the actual N₂ fluxes would be 2.0 kg N ha⁻¹, still leaving a gap of 14 kg N ha⁻¹ for OF and 6 kg N ha⁻¹ for IF. This means that method-inherent underestimation of the N₂-flux cannot close the balance, however we could detect significant fluxes only for approx. two weeks after fertilizer application. Therefore, we were not able to assess the influence of rewetting events on N₂ emissions, which may significantly contribute to N₂ emissions. In addition, some part of the imbalance may be due to underestimation of NH₃ emissions. To take up this suggestion, we revised and rearranged the text to (Lines 459-474):*

"… Despite the amount of $N_2$ losses quantified in this study agreeing with the past studies, the direct field measurement of $N_2$ fluxes in addition to all other relevant loss pathways did not result in a closed $^{15}N$ balance, with the average imbalance being 15 and 7 kg N ha$^{-1}$ for OF and IF, respectively. While some part of the unrecovered $^{15}N$ may be due to an underestimation of $NH_3$ loss, $N_2$ emissions may be underestimated as well (Yankelzon et al., 2024a). Heterogeneous $^{15}N$ distribution in the microplot soil resulting from surface or slit

application of the slurry and $^{15}N_2$ diffusion and storage in subsoil layers contributes to the underestimation of $N_2$ flux rates. These complications associated with $^{15}N$ labelling were discussed in previous studies which indicate that fluxes may be underestimated by up to 30-50% (Vanden Heuvel et al., 1988; Arah, 1997; Well et al., 2018; Well et al., 2019; Friedl et al., 2020; Micucci et al., 2023; Dannenmann et al., 2024). Even assuming a 50% underestimation, $N_2$ emissions after fertilization are in the range of 2 kg N ha$^{-1}$, indicating that the method-inherent potential underestimation alone cannot close the N-balance.

However, the short coverage of the direct $N_2$ measurements, which is due to the short period of time during which the isotopic enrichment of the N pool subject to denitrification is sufficiently high so that the $N_2$ flux can be detected, may be an additional source for underestimation. Consequently, we cannot exclude that additional $^{15}N$ in $N_2$ is emitted in the months following fertilizer application, particularly during rewetting events or towards the end of the growing season when plants decrease their water uptake and water content increases, as observed by Almaraz et al. (2024)."
* * *
10. L 450 "ratio in this study ranged from 0.01 to 1.00" to 1? Is this a mistake? this would mean no N2, only N2O - I think you do not have such case, N2 flux is always much higher than N2O flux

**Answer:** *Thank you for pointing this out. After revising the data, we confirm that the maximum $N_2O/(N_2O + N_2)$ ratio was approximately [0.01 to 0.45], not 1.00. The text has been corrected to reflect this and the figure as well. The sentences now read as (Line 301 and line 480):*

"The ratio of total $N_2O : (N_2+N_2O)$ showed a similar progression over time for both sites, ranging from 0.01 to 0.45."

"The $N_2O : (N_2 + N_2O)$ ratio in this study ranged from 0.01 to 0.45 across both sites using different slurry application techniques….."
* * *
11. L 519 – 522 Did you try to extrapolate these N2 and N2O losses? It is possible to try some extrapolation and asses if this could explain the missing 15N? Eg. Assuming theoretical values of eg. half detection limit for the further period (after these losses can be detected) ? Would this be significant in the N budget change?

**Answer:** *Thank you for this comment. We agree that such extrapolation is useful for $N_2$ as the magnitude of these emissions is relevant for the budget, but cumulative $N_2O$ emissions are much smaller. In addition, please note that we indicate that unrecovered $N_2$ is not the only reason for an imbalance in the N budget ($NH_3$ emission, other uncertainties as explained in subsequent section). For this reason, we revised the manuscript as follows (Lines 550-560):*

"The only measurements that don't cover the whole cultivation period are those of $N_2O$ and $N_2$, suggesting that underestimation of these N losses due to coverage of measurements of only a fraction of the whole cultivation period could explain the unrecovered N losses. Since $N_2O$ emissions are approximately a factor of 10 lower than $N_2$ emissions (Scheer et al., 2020), $N_2$ emissions may have contributed the main part to the unrecovered losses. Assuming a background $N_2$ emission rate of 87 µg m$^{-2}$ h$^{-1}$, which is equivalent to an emission if measured isotope ratios are increased by one standard deviation compared to the background, 0.7, 2.5 and 3.7 kg $N_2$-N ha$^{-1}$ are released during green rye, maize and ryegrass cultivation, respectively. Such a background emission together with slightly underestimated $NH_3$ emissions could explain the unrecovered $N_2$ losses for IF. Overall recovery for OF was close to or lower than that of IF, suggesting that for the OF, additional $N_2$ was emitted during the

cultivation period, which could be due to more frequent denitrification events caused by higher soil bulk density (Table 1, Luo et al., 2000; Hamonts et al., 2013)."
* * *
12. 547 – 549 "reduction of uncertainty for determination of (…), N2 and N2O emission is not in view" - this is not fully true, because there are some ideas of enhancement of the 15NGF for in situ measurements (see https://link.springer.com/article/10.1007/s00374-024-01806-z ) , so that maybe better sensitivity for N2 can be attained, but with large costs and efforts.

*Answer: We sincerely appreciate the reviewer's insightful feedback and for directing us to the recent methodological advancements by Eckei et al. (2025). We agree that the improved 15N gas flux method ($^{15}$NGF+), which combines 15N labelling with helium-oxygen flushing to reduce atmospheric $N_2$ background, represents a significant step toward enhancing sensitivity for $N_2$ detection in field-scale studies. Their work demonstrates that lowering the $N_2$ background to <2% enables more precise quantification of $N_2$ and $N_2O$ fluxes, particularly when paired with production-diffusion modelling to account for subsoil processes.*

*However, as Eckei et al. (2025) emphasize, practical challenges remain, including the high technical complexity, specialized equipment requirements, and labor-intensive protocols (e.g., prolonged gas flushing, isotopic analyses, and model corrections). These constraints currently limit the method's widespread adoption for routine monitoring or large-scale assessments. Consequently, while $^{15}$NGF advances our capacity to study denitrification dynamics, its application remains resource-intensive and context-dependent, with unresolved uncertainties in scaling results across heterogeneous field conditions.*

*We have revised our statement in the manuscript, which now reads as follows (Lines 582-590):*

"Following the same line of argumentation, reduction in the uncertainty of the direct measurement of N balance components, i.e., losses through leaching, $NH_3$, $N_2$ and $N_2O$ could help resolving differences more accurately. While recent advancements, such as the improved $^{15}N$ gas flux method ($^{15}$NGF+) demonstrate potential for enhanced sensitivity in quantifying $N_2$ emissions under field conditions (Eckei et al., 2025), such approaches are expensive and technically challenging which will delay their use in studies targeting complete N balances. Since the same applies to significant reduction of uncertainty in determining leaching losses, $NH_3$ volatilization, and $N_2O$ emissions, it appears like these loss pathways were not markedly influenced by management history given the current measurement frameworks."